# OxPhos defects cause hypermetabolism and reduce lifespan in cells and in patients with mitochondrial diseases

Gabriel Sturm [1,2], Kalpita R. Karan[1], Anna S. Monzel[1], Balaji Santhanam[3], Tanja Taivassalo[4], Céline Bris[5,6], Sarah A. Ware[7], Marissa Cross[1], Atif Towheed[1,8], Albert Higgins-Chen [9], Meagan J. McManus[10,11], Andres Cardenas [12], Jue Lin[2], Elissa S. Epel[13], Shamima Rahman [14], John Vissing[15], Bruno Grassi[16], Morgan Levine [17], Steve Horvath [17], Ronald G. Haller[18], Guy Lenaers[5,6], Douglas C. Wallace [11], Marie-Pierre St-Onge[19], Saeed Tavazoie[3], Vincent Procaccio[5,6], Brett A. Kaufman[7], Erin L. Seifert[20], Michio Hirano[21] & Martin Picard [1,21,22✉]

Patients with primary mitochondrial oxidative phosphorylation (OxPhos) defects present with fatigue and multi-system disorders, are often lean, and die prematurely, but the mechanistic basis for this clinical picture remains unclear. By integrating data from 17 cohorts of patients with mitochondrial diseases ($n = 690$) we find evidence that these disorders increase resting energy expenditure, a state termed *hypermetabolism*. We examine this phenomenon longitudinally in patient-derived fibroblasts from multiple donors. Genetically or pharmacologically disrupting OxPhos approximately doubles cellular energy expenditure. This cell-autonomous state of hypermetabolism occurs despite near-normal OxPhos coupling efficiency, excluding uncoupling as a general mechanism. Instead, hypermetabolism is associated with mitochondrial DNA instability, activation of the integrated stress response (ISR), and increased extracellular secretion of age-related cytokines and metabokines including GDF15. In parallel, OxPhos defects accelerate telomere erosion and epigenetic aging per cell division, consistent with evidence that excess energy expenditure accelerates biological aging. To explore potential mechanisms for these effects, we generate a longitudinal RNASeq and DNA methylation resource dataset, which reveals conserved, energetically demanding, genome-wide recalibrations. Taken together, these findings highlight the need to understand how OxPhos defects influence the energetic cost of living, and the link between hypermetabolism and aging in cells and patients with mitochondrial diseases.

A full list of author affiliations appears at the end of the paper.

Mitochondrial diseases are caused by mutations in either the mitochondrial (mtDNA) or nuclear (nDNA) genomes, which impair oxidative phosphorylation (OxPhos) and the ability to convert food substrates into ATP[1]. However, cellular dysfunction arises even when ATP levels are normal[2–4], suggesting that energy *deficiency* may not be the primary disease initiator. In animal models, OxPhos defects trigger nuclear transcriptional responses, including the integrated stress response (ISR)[3,5–8]. As a result, downstream gene products such as growth differentiation factor 15 (GDF15) are secreted systemically, where they impact organismal metabolic functions[9,10]. This implicates conserved systemic signaling pathways in the pathogenesis of mitochondrial diseases[11]. Considering that these stress pathways entail fundamentally energetically demanding cellular processes (transcription, translation, secretion, etc), OxPhos defects could therefore increase energy consumption at the cellular and organismal levels. However, the potential metabolic costs of cellular and systemic recalibrations in mitochondrial disorders have not been defined.

Clinically, OxPhos defects cause a broad spectrum of multisystem disorders where symptoms include, among others, fatigue and exercise intolerance[12,13]. As a result, most patients with mitochondrial diseases avoid physical activity and exercise[14,15]. A common misconception arising from this clinical picture is that a reduced mitochondrial capacity to oxidize substrates[16] coupled with minimal physical activity levels would promote an energy conservation response, resulting in positive energy balance and body fat accumulation, leading to obesity. However, patients with mitochondrial diseases are rarely obese. In fact, patients with moderate to severe disease, on average, classify as underweight[17]. Although gastrointestinal symptoms that limit food intake or absorption could contribute to this phenotype, the rarity of obesity in mitochondrial disease remains a clinical paradox. This may be resolved by the counterintuitive notion that mitochondrial OxPhos defects may not decrease energy consumption and expenditure but may rather *increase* the energetic cost required to sustain basic physiological functions.

Living organisms avoid thermodynamic decay to grow and survive by consuming energy. The amount of energy expended relative to the minimal metabolic rate required to sustain life is defined as *metabolic efficiency*. Strong evolutionary pressures have optimized metabolic efficiency in organisms, thereby minimizing the amount of ATP required to sustain life[18]. One evolutionary strategy includes the choice of metabolic pathways to derive ATP (OxPhos vs glycolysis), which have different ATP yields and metabolic costs[19]. Within cells, metabolic costs arise mainly from transcription/translation processes (~60% of total energy demands), the maintenance of ionic balance, as well as organelle biogenesis and degradation[20,21], which includes mitochondrial turnover. Consequently, mitochondrial biogenesis entails a substantial energetic cost because of the extensive mitochondrial proteome[19]. In mitochondrial diseases, the intracellular heterogeneous mixture of mitochondria with mutant and wild-type mtDNA (i.e., heteroplasmy) triggers exaggerated biogenesis[22], a phenomenon predicted to increase the basal metabolic cost of organelle maintenance and total energy expenditure[23]. Accordingly, a re-analysis of resting energy expenditure (REE) in animal models of mitochondrial OxPhos defects indicates that REE is likely elevated by 15–85%, including in Crif1$^{-/-}$ mice with impaired mitochondrial translation[10], Clpp$^{-/-}$ mice with deficient proteostasis[24], Polg mutator mice[10], ANT1$^{-/-}$ mice with impaired ATP/ADP exchange[25], and *ATP6*-mutant flies[26]. Based on thermodynamics principles, impaired OxPhos capacity may impede the natural and optimal balance of energy transformation pathways, consequently reducing metabolic efficiency. Therefore, we reasoned that patients with severe OxPhos defects

would similarly exhibit impaired metabolic efficiency and increased REE—a state known as *hypermetabolism*. Other causes of OxPhos defects, including mutations in nuclear genes encoding respiratory chain assembly factors like *SURF1*[27], which cause disease and decrease lifespan in humans[28], could also trigger hypermetabolism.

Shortened lifespan is a ubiquitous feature of mitochondrial diseases[29–31], and most animal models with severe OxPhos defects die prematurely[32–35]. But is there a causal link between hypermetabolism and lifespan in humans? Among healthy individuals, elevated REE or hypermetabolism measured by indirect calorimetry (oxygen consumption, VO$_2$) predicts more rapid age-related physiological decline[36] and independently predicts 25–53% higher mortality over the following 20–40 years[37,38]—an effect double than what is incurred by smoking cigarettes[38]. In human stem cells, hypermetabolism was also correlated with senescence and other aging phenotypes[39]. Mechanistically, multiple processes compete for limited energetic resources within cells[40,41], and also within organisms[42,43], particularly under energy-restricted conditions. Some cellular operations are prioritized over others[20]. As a result, the energetic cost of stress responses[43] and their associated increase in transcription/translation can inhibit growth and cell division, even triggering premature senescence[44,45]. Recently, it was reported that excessive activation of the ISR alone inhibits cell population growth[8]. Thus, OxPhos-induced ISR activation and the resulting hypermetabolism could curtail growth and/or cause premature death by forcing an energetic tradeoff between stress responses and growth/survival pathways.

Taken together, the observations that: (i) genetic mitochondrial OxPhos defects trigger ISRs, (ii) cells operate under energetic constraints where the prioritization of stress responses and transcription/translation costs can precipitate senescence, and (iii) decreased metabolic efficiency predicts shorter lifespan in humans and other animals, lead to the following hypothesis: OxPhos defects trigger hypermetabolism both physiologically and cell-autonomously, a phenotype associated with reduced lifespan.

Here we test this hypothesis first by re-analyzing data from multiple clinical cohorts of primary mitochondrial diseases with direct and indirect assessments of energy expenditure and lifespan, and then via longitudinal in vitro studies in primary human and patient-derived fibroblasts. We have developed a cellular system that provides high temporal resolution, repeated-measures of bioenergetic and multi-omic molecular recalibrations across the cellular lifespan. Using this model, we show that both genetic and pharmacological mitochondrial OxPhos defects trigger marked hypermetabolism in a cell-autonomous manner. We identify mtDNA instability, activation of the ISR, increased secretory activity, and transcriptional upregulation of transcriptional/translational stress pathways as potential contributors to hypermetabolism. Finally, we report that OxPhos defects and hypermetabolism are linked to accelerated telomere shortening and epigenetic aging in fibroblasts, and provide a publicly available longitudinal dataset to query epigenetic and transcriptional signatures conserved across both cellular models. Our analyses highlight how the associated resource dataset can serve as a discovery platform to identify potentially targetable pathways linking OxPhos defects to hypermetabolism, as well as downstream mechanisms linking hypermetabolism to cellular and clinical phenotypes. Together, these translational data implicate hypermetabolism as a pathophysiological feature of mitochondrial diseases and lifespan reduction.

## Results

**Meta-analysis of metabolic rate and physiology in primary mitochondrial diseases**. To test the hypothesis that mitochondrial OxPhos defects are associated with increased energy

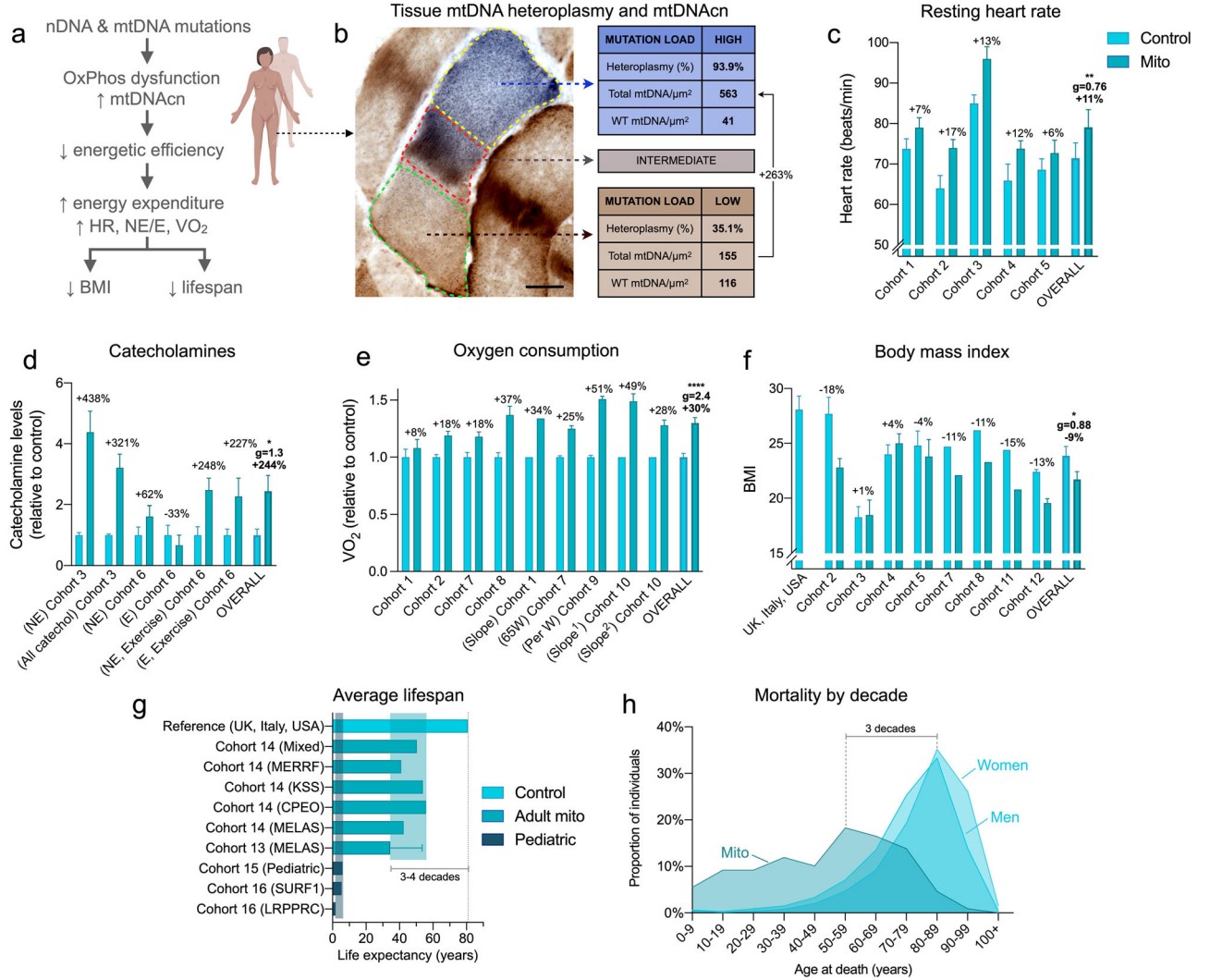

**Fig. 1 Meta-analysis of human studies reveals increased energy expenditure and shortened lifespan in primary mitochondrial diseases. a** Overall conceptual model linking mtDNA- and nDNA-related OxPhos defects to impaired metabolic efficiency at the cellular level, impacting whole-body resting energy expenditure and clinical outcomes. **b** Skeletal muscle biopsy with individual muscle fibers stained with cytochrome c oxidase/succinate dehydrogenase (COX/SDH) histochemistry to reveal functional (brown) and respiratory chain deficient (blue) mitochondria. In the affected cell (middle), three sub-regions showing low, intermediate, and high mtDNA mutation load were captured by laser capture microdissection and subjected to quantitative PCR analysis as in ref. [117]. Subcellular regions with high mtDNA mutation load show elevated mtDNA density, which is predicted to increase the energetic cost due to mitochondrial biogenesis and turnover processes. WT, wild type. **c** Meta-analysis of human mitochondrial disease cohorts showing elevated resting heart rate ($n = 104$ controls, 111 patients), **d** catecholamines (urinary-Cohort 3 and blood-Cohort 6) at rest or during fixed-intensity exercise ($n = 38$ controls, 19 patients), **e** whole-body oxygen consumption measured by indirect calorimetry at rest or during the response to mild exercise challenge; one before training, two after training. Slope refers to the rate of increase in $VO_2$ relative to work rate, where a higher slope indicates increased energetic cost for a given work rate ($n = 56$ controls, 78 patients). **f** Body mass index (BMI) across mitochondrial disease cohorts and compared to relevant national averages (USA, UK, and Italy combined) ($n = 285$ controls, 174 patients). **g** Average life expectancy in individuals with mitochondrial diseases relative to the national average ($n = 301$ patients). Data are means ± SEM, with % difference between mitochondrial disease and control group were available. **h** Mortality (age at death) over 10 years (2010–2020) in Cohort 17 compared to national averages for women and men ($n = 109$ patients). See Table 1 for cohort details. Total $n = 225$ healthy controls, 690 patients. Only aggregate group means (with or without a measure of variance) were available for some cohorts, so individual participant data is not shown. Standardized effect sizes are quantified as Hedges' g (**g**). Overall group comparisons were performed by paired t tests (**c** and **f**) or one-sample t tests (**d** and **e**), *$p < 0.05$, **$p < 0.01$, ****$p < 0.0001$.

expenditure and shortened lifespan (Fig. 1a), we integrated and re-analyzed data from a total of 17 cohorts representing a total of 690 patients with mitochondrial diseases and 225 healthy controls (provided by the authors or directly from publications) (Table 1). The heterogeneous mixture of functional and dysfunctional mitochondria within single cells is well-known to cause mitochondrial hyperproliferation and increase mtDNA copy number[46–49], as illustrated within a single patient skeletal muscle cell in Fig. 1b. Increased biogenesis must naturally incur increased

energy expenditure at the cellular level[23], which we reasoned may translate to elevated whole-body REE.

In patients with mitochondrial diseases, resting heart rate, which correlates with whole-body REE[50], was on average 10.7% higher than healthy controls ($p < 0.01$, Fig. 1c). This tachycardia reached up to +46% when patients and controls performed mild exercise at the same absolute workload. Both at rest and during mild physical activity, as initially reported in a small study[51], patients had on average 244% higher blood or urine catecholamine levels ($p < 0.05$,

**Table 1 Human cohorts included in the quantitative meta-analysis of energy expenditure and related clinical phenotypes in patients with mitochondrial diseases (see Fig. 1).**

| Cohort # | Author (year) | N = (W/M) | Age | Genetics | Mutations | Clinical |
|---|---|---|---|---|---|---|
| Cohort 1 | Taivassalo (2003) | 40 Mito (22/18); 32 Ctrl (9/22) | 37; 39 | mtDNA (n = 35) nDNA (n = 5) | m.3243 A > G, m.8344 A > G, m.14710 G > A, m.5543 T > C, m.4409 T > C, m.14846 G > A, m.5920 G > A, ND2 and COXIII microdeletions, sDel, mDel, other (3), unknown (4) | CPEO, MELAS, MERRF, EI, mixed |
| Cohort 2 | 5a:Bates (2013) 5b:Newman (2015) 5c:Galna (2014) 5d: Gorman et al. Newcastle cohort | a:10 Mito (4/6); 10 Ctrl (4/6) b:8 Mito (5/3) c:6 Mito (1/5) d:8 Mito (2/6) | a:42.4; 39.0 b:42 c:40.5 d:42 | mtDNA | a:m.3243 A > G b:m.3243 A > G c:m.8344 A > G + 3243 A > G d:sDel | SNHL, DM, Ei, AT, FT, DP, mixed |
| Cohort 3 | Strauss (2013) | 9 Mito (7/2); 28 Ctrl | 14.6; 14.0 | nDNA | SLC25A4 (ANT1) mutations (c.523delC, p.Q175RfsX38) | CM, EI, insomnia, DP, anxiety |
| Cohort 4 | Delaney (2017) | 21 Mito (15/6); 12 Ctrl (8/4) | 44; 34 | mtDNA | sDel, mDel, m.3243 A > G, m.10010 T > C, m.12261 T > C, ISCU, m.4281 A > G, CYTB, m.8344 A > G, m.5543 T > C | Mild to severe mixed |
| Cohort 5 | MiSBIE | 23 Ctrl (15/8); 12 Mito (8/4) | 34.0; 32.9 | mtDNA | m.3243 A > G | MELAS, mixed |
| Cohort 6 | Jeppesen (2013) | 10 Mito (6/4); 10 Ctrl (6/4) | 39; 39 | mtDNA | m.3243 A > G, 8344 A > T, 4409 T > C, 8340 G > A, 2-bp deletion, 12,113–14422, 7177–13767 | CPEO, EI, HI, GI, Enc, SS, DM, ME, AT |
| Cohort 7 | Jeppesen (2009) | 10 Mito (5/5); 10 Ctrl (5/5) | 39; 40 | mtDNA | m.3243 A > G, m.8344 A > T, m.5543t > C, sDel | CPEO, EI, HI, GI, Enc, SS, DM, ME, AT |
| Cohort 8 | Heinicke (2011) | 5 Mito (2/3); 4 Ctrl (2/2) | 42; 34 | mtDNA, nDNA | m.3243 A > G, m.5543 T > C, m.14846 G > A, ISCU | Myopathy |
| Cohort 9 | Grassi (2009) | 15 Mito (7/8); 21 PCtrl[a] (7/14); 22 Ctrl (9/13) | 40.1; 38.3; 37.9 | mtDNA | sDel, mDel, m.8344 A > G | Myopathy |
| Cohort 10 | Porcelli (2016) | 6 Mito (2/4) | 51 | mtDNA | mDel, sDel, m.3255 G > A, m.3243 A > G | Myopathy |
| Cohort 11 | Grassi (2007) | 6 Mito (1/5); 25 PCtrl[a] (5/20); 20 Ctrl (8/12) | 37.8; 31.6; 32.7 | mtDNA | mDel, m.8344 A > G | Myopathy |
| Cohort 12 | Hou (2019) | 89 Mito (57/32) | 30.4 | mtDNA, nDNA | sDel, POLG, RRM2B, Twinkle, TK2, m.3243 A > G, m.8344 A > G, m.5541 C > T, m.10158 C > T | MELAS, CPEO |
| Cohort 13 | Kaufman (2011) | 31 Mito (16/15); 54 Ctrl[b] (15/39) | 30; 38 | mtDNA | m.3243 A > G | MELAS |
| Cohort 14 | Barends (2015) | 30 Mito (15/15) | 50.4[c] | mtDNA, nDNA | m.3243 A > G, sDel, mDel, c.1635C > G, m.8344 A > G, m.13094 T > C, m.14709 T > C, m.5816 A > G, m.14484 T > C, m.12258 G > A, POLG mutations | MELAS, CPEO, KSS, MERRF, mixed |
| Cohort 15 | Eom (2017) | 221 Mito Pediatric | 6.0[c] | mtDNA, nDNA | m.3243 A > G, LS mutations | LS, MELAS, mixed |
| Cohort 16 | Wedatilake (2013) | 44 Mito (20/24) Pediatric | <14[c] | nDNA | SURF1 mutations | Poor feeding/ vomiting, PWG, DD, HT, MD, AT |
| Cohort 17 | McFarland et al. Newcastle cohort | 109 Mito (56/53) | 48.1 | mtDNA, nDNA | In addition to Cohort 14: AGK, ETFDH, m.10010 T > C, m.11778 G > A, m.13513 G > A, m.8993 T > C, m.8993 T > G, m.9176 T > C, m.9997 T > C, MRPL44, NDUFAF6, NDUFS1, RRM2B, SDHA, SURF1, TYMP | MELAS, PMM, MERRF, MIDD, MNGIE, KSS, CM, mixed |

ANT1 adenine nucleotide translocator 1, AT ataxia, CM cardiomyopathy, CPEO chronic progressive external ophthalmoplegia, DD developmental delay, Dm diabetes mellitus, DP depression, EI pure exercise intolerance, Enc encephalopathy, FT fatigue, GI glucose intolerance, HI hearing impairment, HT hypotonia, KSS Kearns-Sayre Syndrome, LS Leigh Syndrome, MD movement disorder, mDel multiple mtDNA deletions, ME myoclonic epilepsy, MELAS mitochondrial encephalopathy, lactic acidosis, stroke-like episodes, MERRF myoclonus epilepsy with ragged red fibers, MiSBIE mitochondrial stress, brain imaging, and epigenetics study, mtDNA mitochondrial DNA, nDNA nuclear DNA, PWG poor weight gain, sDel single, large-scale mtDNA deletion, SNHL sensorineural hearing loss, SS: short stature.
[a]PCtrl: "patient controls" with symptoms of mitochondrial myopathy but with biopsy negative for primary mitochondrial disease.
[b]Controls were m.3243 A > G carrier relatives without MELAS.
[c]Based on age at death. The number of women (W) and men (M) are shown in parentheses.

Fig. 1d), particularly norepinephrine (NE), a neurohormone sufficient to elevate REE when administered systemically to healthy individuals[52]. To estimate REE in mitochondrial disease patients, we used resting whole-body $VO_2$ expressed relative to body weight, which, although imperfect, was available in the largest number of studies. Strikingly, $VO_2$ measured by indirect calorimetry across 6 cohorts of patients with mtDNA defects was on average 30% higher at rest ($p < 0.0001$) than in healthy controls, a difference characterized by very large effect size (Hedge's $g = 2.4$, Fig. 1e). REE estimates using the Weir equation[53] (combining both $VO_2$ and $VCO_2$, readily available in 3/6 cohorts) yielded equivalent results within 1.2% of the group difference derived from $VO_2$ alone. Notably, $VO_2$ was elevated by more than half (+51%) during mild physical activity in mitochondrial diseases, consistent with hyperkinetic cardiocirculatory responses to exercise in this population[54]. Thus, these gross body mass-normalized REE values reveal increased energy consumption (i.e., lower metabolic efficiency) in mitochondrial diseases, at rest and particularly during mild physical challenges.

The increase in REE is particularly striking given that patients with mitochondrial diseases, on average, have lower muscle mass[17], which is the major site of activity-dependent energy consumption. The lower muscle mass in patients would be expected to reduce energy expenditure, unless the tissues intrinsically exhibited impaired metabolic efficiency, and thus consumed more energy per unit of time just to sustain homeostasis. Therefore, the meta-analysis of these clinical data from multiple cohorts combining hundreds of patients reveals an increased energetic cost of living per unit of body mass – or *hypermetabolism*—in mitochondrial diseases.

Physiologically, hypermetabolism is expected to produce a negative energy balance, expending more energy substrates than are ingested, generally preventing the accumulation of body fat. Accordingly, body mass index (BMI), a gross estimate of adiposity, was on average 9.8% lower ($p < 0.05$) in patients with mitochondrial diseases compared to controls (23% lower than national averages across 3 countries) (Fig. 1f). In one study, fat mass index, a more precise indicator of body fat, was 21.9% lower in mitochondrial

disease patients[17]. Moreover, although not all patients are thin, patients with more severe disease manifestations tended to have lower BMI ($r = -0.25$, $p = 0.018$)[17], suggesting that more severe mitochondrial OxPhos defects in humans compete with body fat accumulation and obesity. Again, this result is in line with those in animal models of OxPhos defects, which exhibit hypermetabolism and reduced adiposity[10,24–26].

This clinical picture of mitochondrial diseases marked by increased REE and reduced body fat was associated with a 3–4-decade reduction in lifespan among adults (Fig. 1g)[31]. In a 10-year longitudinal observational study from the UK Newcastle group, peak mortality in mixed genetic diagnoses of mitochondrial diseases occurs up to 3 decades earlier than the national reference (Fig. 1h). In children with severe pediatric forms of mitochondrial diseases, including diseases caused by autosomal recessive respiratory chain defects (e.g., SURF1 mutations: median lifespan 5.4 years[28]), lifespan can be reduced by >90%. Heterogeneity between genetic diagnoses also highlights possible mutation-specific effects on hypermetabolism (Supplementary Fig. 1). Together, these multimodal physiological data establish hypermetabolism as a clinical feature of mitochondrial diseases, which could account for the rarity of obesity and possibly also contribute to the shortened lifespan in this population.

**Longitudinal analysis of primary human fibroblasts with SURF1 mutations.** To examine if mitochondrial OxPhos defects alter the REE and lifespan in a cell-autonomous manner independent of clinical, medical, and socio-behavioral confounds, we next performed a longitudinal study of primary human fibroblasts with genetically defined or pharmacologically induced OxPhos defects. We used cells with a stable nuclear mutation in SURF1 (Surfeit Locus Protein 1), which causes partial mis-assembly and dysfunction of respiratory chain complex IV (cytochrome c oxidase, COX)[55], leading to Leigh syndrome and death in early childhood (see Fig. 1g). Primary dermal fibroblasts were obtained from 3 patients with SURF1 mutations presenting with Leigh syndrome, and from 3 healthy donors with no known mitochondrial defects (Control). Each group included one female and two male donors. To capture both baselines as well as trajectories of metabolic parameters across the entire lifespan, we passaged each fibroblast line over multiple cellular generations until growth arrest, a model that recapitulates in vivo molecular features of human aging, including canonical age-related changes in telomere length[56] and DNA methylation[57]. By sampling cells across the lifespan, longitudinal profiles of multiple cellular, bioenergetic, transcriptomic, epigenomic, and secreted molecular features can be modeled for each donor (Fig. 2a). Although healthy cells survive for up to 250 days, here we limit our analyses to the maximal lifespan of SURF1-mutant cells, ~150 days.

Beyond allowing longitudinal assessments of molecular and bioenergetic parameters as cells transition from early-, mid-, and late-life, one major advantage of time-resolved trajectories with repeated measures is that this approach de-emphasizes potential bias of any single time point and provides more accurate estimates of stable cellular phenotypes for each donor and treatment condition. The use of primary human cells obtained from multiple donors, compared to the same experiment repeated in an immortalized cell line(s), also provides a more robust test of the generalizability of the data. Throughout the text, we report standardized measures of effect sizes (Hedge's g) where $g > 0.2$ is considered a small, $g > 0.5$ a medium, and $g > 0.8$ represents a large effect size, which is considerably more informative than $p$ values to compare groups with small sample size (3 donors per group)[58].

**SURF1 mutations cause hypermetabolism.** We first examined the effect of SURF1 mutations using extracellular flux analysis

(Seahorse XF$^e$96) of oxygen consumption rate (OCR) and extracellular acidification rate (ECAR) (Fig. 2b, c). Using standard stoichiometric ratios for oxygen consumed, protons pumped, and linked ATP synthesis under standard conditions, OCR and ECAR can be transformed into interpretable ATP production rates using the methods described in ref. 59. When added together, OxPhos-derived ($J_{ATP-OxPhos}$) and Glycolysis-derived ATP flux ($J_{ATP-Glyc}$) reflects the total energetic demand ($J_{ATP-Total}$) of each cell population (Supplementary Figure 2a). This approach is the cellular equivalent to REE measurements through indirect calorimetry in humans (Fig. 1e) and mice[10,24,25].

Trajectories of $J_{ATP-OxPhos}$ and $J_{ATP-Glyc}$ across 150 days of lifespan are presented in Fig. 2d. As expected from the SURF1 deficiency, SURF1-mutant cells (hereafter SURF1 cells) exhibited a 44% decreased $J_{ATP-OxPhos}$, but a 3-4-fold increased $J_{ATP-Glyc}$. Both parameters remained relatively stable across the lifespan (although a potential oscillatory behavior cannot be ruled out). Computing total energy expenditure showed that total ATP demand per unit of time was strikingly 91% higher in SURF1 cells relative to control cells ($J_{ATP-Total}$, $p < 0.001$, $g = 2.4$) (Fig. 2e). These data demonstrate a robust SURF1-induced hypermetabolic state similar, albeit of greater magnitude, to that observed in patients with mitochondrial diseases.

To confirm this finding, potential confounds had to be ruled out. Non-glycolytic ECAR could inflate estimates of $J_{ATP-Glyc}$[60]. However, measured resting non-glycolytic ECAR (in the absence of glucose or in the presence of the glycolysis inhibitor 2-deoxyglucose) was not elevated in SURF1 cells (it was, in fact, 31% lower), confirming the specificity of the ECAR signal in SURF1 cells to glycolysis (Supplementary Figure 3d). We also confirmed that non-OxPhos-related oxygen consumption by cytoplasmic and other oxidases did not differ between experimental groups (Supplementary Fig. 2d). Non-mitochondrial respiration also is not included in computing $J_{ATP-OxPhos}$, which formally excludes this parameter as a potential contributor to the hypermetabolism measured in SURF1 cells.

Primary fibroblasts are continually dividing, and a portion of the total energy budget is expected to support cell division-related processes, including DNA replication, transcription/translation, and other intracellular processes[20]. Early in life (20–50 days), when division rates were mostly constant, SURF1 fibroblasts compared to control cells divided on average 31.8% *slower* ($p < 0.0001$, $g = -1.53$; and 48.4% slower when quantified across 150 days). Therefore, hypermetabolism in SURF1 cells cannot be accounted for by an accelerated division rate. In fact, normalizing $J_{ATP-Total}$ per rate of division further exaggerates apparent hypermetabolism, where SURF1 cells expend more than double the amount of energy than controls to complete each cell cycle. Moreover, optically monitoring cell size at each passage showed that the SURF1 cell volume was moderately larger in early life and became smaller with increasing age, reaching a similar volume as control cells by 150 days (Supplementary Fig. 4a). Cell death was not significantly elevated ($p = 0.69$, $g = 0.15$, Supplementary Fig. 4d, e). After accounting for cell volume, energy expenditure remained significantly elevated in SURF1 fibroblasts ($p < 0.0001$, $g = 1.2$, Fig. 2f), demonstrating an increase in volume-specific REE. This increase is consistent in magnitude with that observed in humans (Fig. 1) and animals[10,24–26] with OxPhos defects.

In control cells, the balance of estimated ATP derived from OxPhos and glycolysis was 64:36%, such that under our specific tissue culture conditions (physiological 5.5 mM glucose, with glutamine, pyruvate, and fatty acids), healthy fibroblasts derived the majority of ATP from OxPhos. In contrast, SURF1 deficiency robustly shifted the relative OxPhos:Glycolysis contribution to 23:77% ($p = 4.1e-6$, $g = -5.1$), reflecting a significant shift in OxPhos-deficient cells towards an alternative, and therefore less energy efficient, metabolic strategy (Fig. 2g, h). As expected,

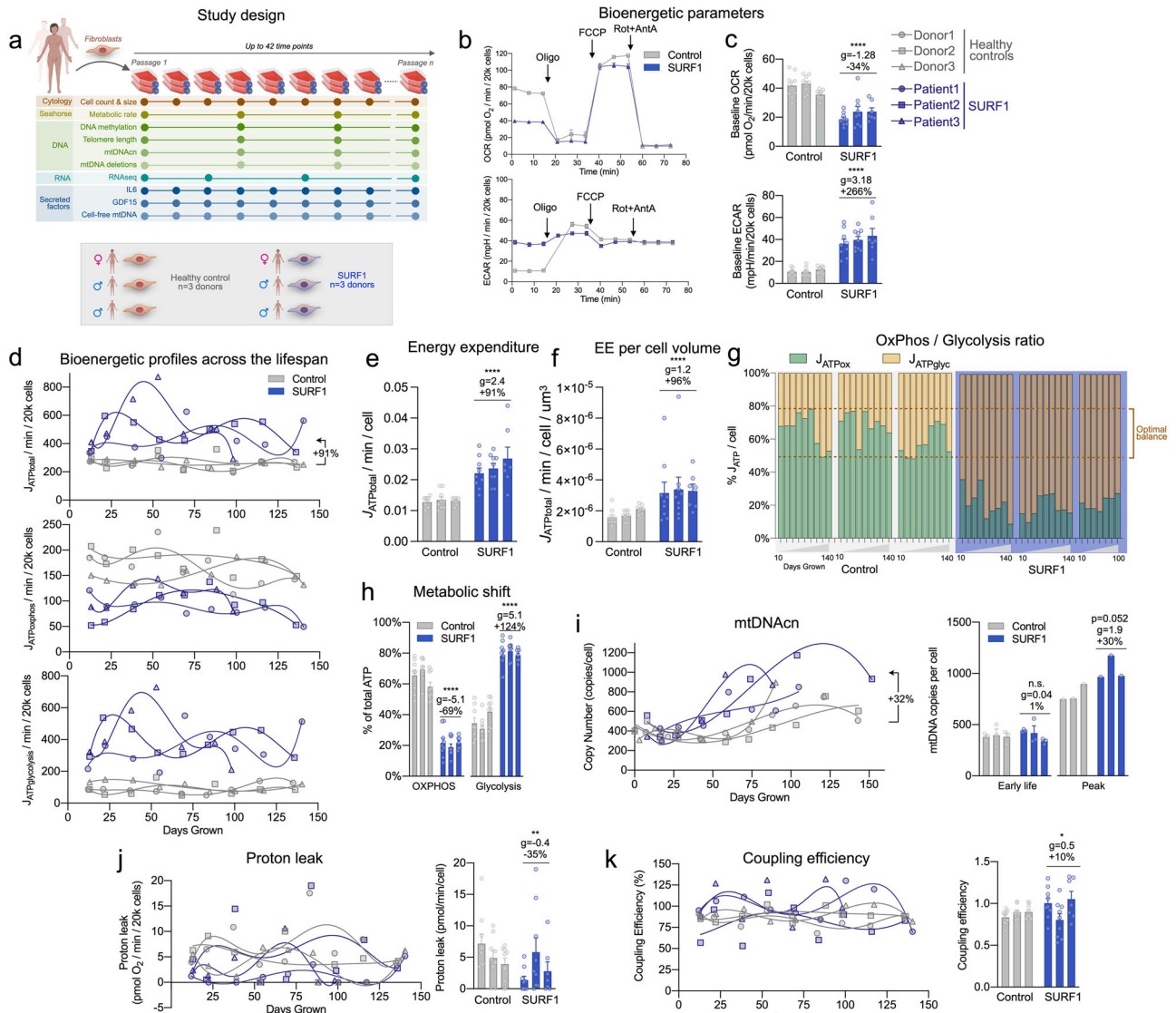

**Fig. 2 SURF1 defects decrease metabolic efficiency and cause hypermetabolism without affecting coupling efficiency. a** Schematic of the study design with primary human fibroblasts, coupled with repeated, longitudinal measures of cellular, bioenergetic, and molecular profiling across the lifespan. Three Control and three SURF1 donors (one female, two males in each group) were used for all experiments. **b** Example oxygen consumption rate (OCR) and extracellular acidification rate (ECAR) obtained from Seahorse measurements of Control and SURF1 cells. **c** Comparison of average OCR and ECAR values across the cellular lifespan. The specificity of the ECAR signal for glycolysis was verified (see *Methods* for details). **d** Lifespan trajectories of ATP production rates ($J_{ATP}$) derived from glycolysis ($J_{ATP-Glyc}$), oxidative phosphorylation ($J_{ATP-OxPhos}$), and total ATP ($J_{ATP-Total}$: Glycolytic- + OxPhos-derived rates) over up to 150 days. Percentages show the average difference between SURF1 and Control across the lifespan. **e** Lifespan average energy expenditure (EE) by cell line and **f** corrected for cell volume. **g** Balance of $J_{ATP}$ derived from OxPhos and glycolysis and **h** quantified SURF1-induced metabolic shift. Dotted lines in (**h**) denote the range in control cells. **i** Lifespan trajectory of mtDNAcn and average mtDNAcn at the first 3 time points (early life, days 5–40) and peak value across the lifespan. **j** Lifespan trajectories and averages of proton leak and **k** coupling efficiency estimated from extracellular flux measurements of ATP-coupled and uncoupled respiration. $n = 3$ individuals per group, 7–9 timepoints per individual. Data are means ± SEM. *$P < 0.05$, **$P < 0.01$, ***$P < 0.001$, ****$P < 0.0001$, mixed effects model (fixed effect of control/SURF1 group and days grown, random effects of the donor or cell line).

removing glucose from the media did not substantially affect growth in control cells, but the absence of glucose was lethal to SURF1 cells within 5 days, confirming their dependency on glycolysis for survival (Supplementary Fig. 3).

In response to this metabolic shift towards glycolysis, we expected SURF1 cells to naturally decrease maintenance-related energetic costs by decreasing mitochondrial mass and mtDNA copy number (mtDNAcn). However, in early life, SURF1 cells had the same mtDNAcn as control cells (5–40 days: $p = 0.99$, $g = 0.04$). And across the lifespan, SURF1 cells contained 32% more mtDNA copies, which manifested as an earlier age-related

rise in mtDNAcn that reached maximal levels on average 30% higher than control cells ($p = 0.52$, $g = 1.9$, Fig. 2i). Thus, although total mitochondrial mass was not directly assessed, elevated mtDNAcn similar to that observed in patient tissues (see Fig. 1b) could contribute to increased maintenance cost and overall hypermetabolism in OxPhos-deficient cells, as suggested by mathematical modeling studies[23].

One potential mechanism for the lowered metabolic efficiency is a decrease in OxPhos coupling (i.e., uncoupling) at the inner mitochondrial membrane. However, both estimated proton leak (Fig. 2j) and coupling efficiency (Fig. 2k) measured by the

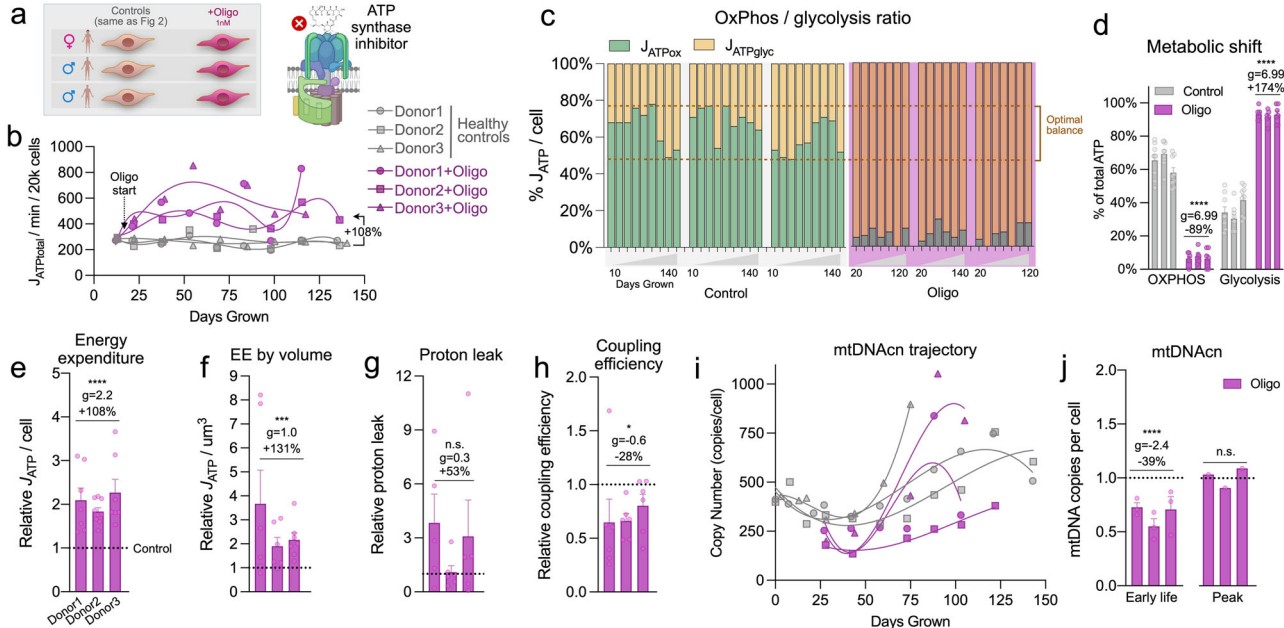

**Fig. 3 Pharmacological inhibition of the mitochondrial $F_oF_1$ ATP synthase triggers hypermetabolism. a** Schematic of the study design for fibroblast profiling across the lifespan from three Control donors treated with 1 nM Oligomycin (Oligo). **b** Lifespan trajectories of $J_{ATP}$ (Glycolytic + OxPhos) derived from oxygen consumption rate (OCR) and extracellular acidification rate (ECAR) obtained from Seahorse measurements across the cells' lifespan (up to 150 days). Percentages show the total average difference between Oligo and Control. **c** Balance of $J_{ATP}$ derived from OxPhos and glycolysis across the lifespan and **d** oligo-induced metabolic shift. Dotted lines denote the range in control cells. **e** Relative average lifespan energy expenditure by cell line normalized to control, **f** corrected for cell volume. **g** Average of proton leak and **h** coupling efficiency measures on the Seahorse normalized to control. **i** Lifespan trajectories and **j** average mtDNA copy number at the first three time points (early life) and peak value across the lifespan. $n = 3$ individuals per group, 7–9 timepoints per individual. Data are means ± SEM. *$P < 0.05$, **$P < 0.01$, ***$P < 0.001$, ****$P < 0.0001$, mixed effects model for Oligo vs. control.

proportion of OxPhos-dependent respiration not linked to ATP synthesis were not different between control and SURF1 groups. These parameters also did not show measurable drift across the lifespan, thus ruling out mitochondrial uncoupling as a mechanism for hypermetabolism.

Finally, oxygen tension can have a marked effect on the metabolism and replicative lifespan of cultured fibroblasts[61], and chronic hypoxia improves survival in fibroblasts with complex I defect and the Ndufs4 mouse model of Leigh syndrome[62]. We, therefore, repeated longitudinal experiments in SURF1 cells at low (3%) $O_2$ in parallel with atmospheric (~21%) $O_2$ (Supplementary Fig. 5a). Compared to 21% $O_2$, the low oxygen condition did not improve population doubling rates (Supplementary Figure 5b, c), nor did it correct or alter hypermetabolism (Supplementary Fig. 5d, e). Results of the low $O_2$ "hypoxia" experiments, as well as the full lifespan aging trajectory of control cells beyond 150 days, are available in the resource dataset (see Data availability statement).

**Inhibition of the mitochondrial $F_oF_1$ ATP synthase triggers hypermetabolism.** Next, to test if hypermetabolism manifests specifically in SURF1 cells or whether it is a more general feature of mitochondrial OxPhos defects, we used an orthogonal pharmacological approach to chronically perturb OxPhos and repeated the lifespan assessments of energy metabolism. Starting at day 20, fibroblasts from the same three healthy donors as above were treated chronically with a sublethal concentration of the mitochondrial ATP synthesis inhibitor Oligomycin (Oligo, 1 nM), which induces the ISR[3,63] (Fig. 3a). Oligo reduced cellular oxygen consumption rate by ~90% while largely maintaining viability, reflected in only a moderate elevation in cell death over time (2.7% in Oligo-treated cells vs. 1.4% in control cells, 20–50 days: $p = 0.078$, $g = 0.70$) (Supplementary Figs. 4d and 6b).

In relation to energy expenditure, Oligo doubled $J_{ATP\text{-}Total}$ across the lifespan for each of the three healthy donors (+108%, $p = 5.9e-9$, $g = 2.2$), recapitulating the hypermetabolic state observed in SURF1 cells (Fig. 3b). This robust elevation in cellular energy expenditure was already evident by 5 days of treatment and remained relatively stable across the lifespan, indicating the rapidity and stability of the adaptive hypermetabolic state. As in SURF1 cells, the hypermetabolic state in Oligo-treated cells was attributable to a markedly increased $J_{ATP\text{-}Glyc}$ in excess of the decline in $J_{ATP\text{-}OxPhos}$, resulting in a shift outside of the optimal (i.e., normal) window of the OxPhos:Glycolysis ratio for these primary human cells grown under physiological glucose concentration (Fig. 3c, d).

Reductions in cell size and division rates are strategies to minimize energetic costs. Oligo caused a small but stable 4.8% decrease in cell size ($p < 0.001$, $g = -0.35$), and decreased cell division rates by 39.1% (days 20–50: $p = 1.3e-5$, $g = -1.31$; 49.6% slower across 150 days) (Supplementary Fig. 4). Taking cell size into consideration showed that Oligo increased energy expenditure per unit of cell volume by 131% ($p < 0.001$, $g = 0.97$) (Fig. 3e, f). Here also, hypermetabolism was not driven by a significant increase in estimated proton leak ($p = 0.19$, $g = 0.27$) (Fig. 3g), although we observed a 34.4% reduction in estimated coupling efficiency ($p < 0.05$, $g = -0.59$) (Fig. 3h), likely arising from the expected elevation in membrane potential from ATP synthase inhibition. Unlike *SURF1* mutations, Oligo decreased mtDNAcn by 39.0% early in life (20–50 days: $p = 3.1e-5$, $g = -2.42$), which subsequently normalized; peak levels were similar to control levels (Fig. 3i, j).

Monitoring weekly the influence of Oligo on cell morphology also revealed an unexpected morphological phenotype. Oligo-treated cells developed into a reticular network, which involved contraction of the cell body and extension of multiple cellular

appendages reminiscent of neuronal dendrites (Supplementary Fig. 6). This reversible phenotype exhibited regular oscillatory behavior (1-week normal morphology, 1-week reticular formation). We note that oscillatory behaviors are naturally energy-dependent[64], and that such dramatic and repeated changes in cell morphology must necessarily involve the remodeling of cell membranes and cytoskeleton through the energy-dependent motor and cytoskeletal components. This morphological phenotype unique to the Oligo treatment could contribute to the higher energy expenditure in Oligo-treated cells (+131% ATP consumption per unit of cell volume) vs. SURF1 cells (+91%), which did not exhibit transitory morphological changes.

**OxPhos defects trigger the ISR and mtDNA instability**. To understand the specific organelle-wide mitochondrial recalibrations in hypermetabolic SURF1 and Oligo-treated cells, we performed bulk RNA sequencing across the lifespan in each donor cell line (average of ~7 timepoints per cell line). We then systematically queried mitochondrial pathways from MitoCarta 3.0[65], in addition to all mtDNA-encoded transcripts (37 genes), and core ISR-related genes (average of ATF4, ATF5, CHOP/DDIT3, and GDF15). Both *SURF1* defects and Oligo treatment downregulated the majority of intrinsic mitochondrial pathways, including mtDNA stability and decay, which was downregulated in both SURF1 ($-15\%$, $p = 1.7e-8$, $g = -1.65$) and Oligo-treated cells ($-19\%$, $p < 0.001$, $g = -0.57$) relative to control (Fig. 4a). Although SURF1 and Oligo-treated cells exhibited similar overall mitochondrial transcriptional changes, some pathways showed opposite responses (e.g., expression of mtDNA-encoded genes, Pathway 3 in Fig. 4b), suggesting the existence of partially specific mitochondrial recalibrations among SURF1 and Oligo models. In hierarchical clustering analysis across all pathways, the ISR pathway diverged most strongly from other pathways, and was upregulated +110% in SURF1 ($p = 6.5e-7$, $g = 1.76$) and +217% in Oligo-treated cells ($p = 1.2e-8$, $g = 0.99$), reaching up to a 16-fold elevation relative to the average of the young healthy donor cells (Fig. 4b). Thus, both models of OxPhos defects and hypermetabolism were associated with upregulation of the ISR, and downregulation of most mitochondrial pathways, notably mtDNA maintenance, suggesting a potential effect on mtDNA stability.

To our knowledge, neither *SURF1* mutations nor Oligo treatment is established to cause mtDNA instability, but given the transcriptional changes described above and that heteroplasmy among mtDNA species is predicted to increase energetic maintenance costs[23], we directly examined mtDNA stability using two approaches. We first used gel electrophoresis on long-range PCR products at multiple time points across the lifespan of control, SURF1, and Oligo-treated cells, then validated the presence of mtDNA deletions across the lifespan by mtDNA sequencing and quantified mtDNA deletion burden using eKLIPse[66] (Fig. 4c, d). Circos plots in Fig. 4d show the breakpoints and heteroplasmy level for each mtDNA deletion (or duplication), at early and late time points along the cellular lifespan. Circos plots for all timepoints investigated (4–14 timepoints per condition) are presented in Supplementary Fig. 7.

Consistent with previous work, healthy fibroblasts do not accumulate appreciable heteroplasmy levels of mtDNA deletions in culture. However, SURF1 cells contained, on average, 17-fold more unique mtDNA deletions than control cells ($p < 0.01$, $g = 1.38$), reaching up to 126 unique deletions at a given time point (Fig. 4e, f). The effect of Oligo treatment was more modest but reached levels 3-fold higher than untreated cells ($p < 0.01$, $g = 0.79$), and up to 20 unique deletions per time point. The majority of deletions eliminated segments of the minor arc and

were, on average, 6.8–7.3 kb in length; deletion size was similar among the three groups (Supplementary Fig. 8a–c). Point mutations were not significantly elevated in SURF1 and Oligo-treated cells, suggesting specificity of mtDNA instability to deletions in this OxPhos-deficient, hypermetabolic state (Supplementary Fig. 8d). Compared to controls where the maximal heteroplasmy levels were 0.13%, SURF1 and Oligo accumulated individual deletions reaching up to 0.40% and 0.19% heteroplasmy among the cell population (Supplementary Fig. 8e, f), which remains low but similar to that observed with aging in human blood and brain tissues[67,68], and possibly noteworthy for replicating fibroblasts.

**SURF1 mutations increase aging-related secretory activity**. We next investigated the outputs of the ISR, including the production of metabokines and cytokines. To broadly characterize changes in the cytokine stress response in patient-derived SURF1 cells across the lifespan, we designed a custom Luminex array targeting age-related proteins identified by plasma proteomics to be upregulated with human aging[69] (Fig. 5). Compared to healthy donors, hypermetabolic SURF1 cells secreted higher levels of cytokines on a per-cell basis, including several pro-inflammatory cytokines, chemokines, and proteoglycans associated with the senescence-associated secretory phenotype (SASP)[70] (Fig. 5a). Of the 27 cytokines detected in extracellular media, SURF1 cells achieved the highest cytokine concentration across the lifespan for 23 (85%) of the cytokines, reaching up to 10-fold higher concentration than a control for one of the cytokines (insulin-like growth factor binding protein, IGFbp-rp1) (Fig. 5b). Upregulated cytokines also included the canonical pro-inflammatory cytokines IL-6 and IL-8. The metabokine GDF15, which is elevated in both mitochondrial disease[71,72] and human aging[69,73], and which also appears sufficient to trigger hypermetabolism in mice[10], was also upregulated by 110% in SURF1 vs. control cells (20–80 days, $p = 0.035$, $g = 1.0$, Fig. 5c).

We attempted to validate IL-6 and GDF15 levels in both SURF1 and Oligo-treated cells by enzyme-linked immunosorbent assays (ELISA). The ELISA confirmed that IL-6 increased exponentially in aging fibroblasts, displaying altered onset and trajectories in both SURF1 (upregulated) and Oligo-treated cells (downregulated) (Fig. 5d). Compared to control fibroblasts where GDF15 was undetectable in early passages, SURF1 mutant fibroblasts began to secrete GDF15 prematurely, and Oligo treatment acutely induced robust GDF15 secretion by 1–2 orders of magnitude over the first few weeks (Fig. 5e), consistent with the rapid induction of the ISR particularly in Oligo-treated cells (see Fig. 4b).

As cell-free mitochondrial DNA (cf-mtDNA) is associated with human aging[74] and was recently found to be elevated in the plasma of patients with mtDNA mutations/deletions[75], we quantified cf-mtDNA in the media along the lifespan. Both mtDNA and nDNA were detectable at appreciable levels (Supplementary Fig. 9a, b). Compared to the media of control cells, cf-mtDNA levels were 73% higher in SURF1 ($g = 0.5$) and 100% higher ($g = 0.3$) in the media of Oligo-treated cells (Fig. 5f), although these differences did not reach statistical significance due to the high temporal variation of this phenotype. Parallel measurements of cell-free nuclear DNA (cf-nDNA) showed that the released mitochondrial-to-nuclear genome ratio was, on average, 117% higher in SURF1 than in control cells ($p < 0.01$, $g = 0.85$, Supplementary Fig. 9c, d), indicative of selective mtDNA release. Given the energetic cost associated with protein secretion[21,76], we suggest that the cytokine/metabokine and mtDNA hypersecretory phenotype in SURF1 and Oligo cells must contribute to hypermetabolism along with other cellular processes.

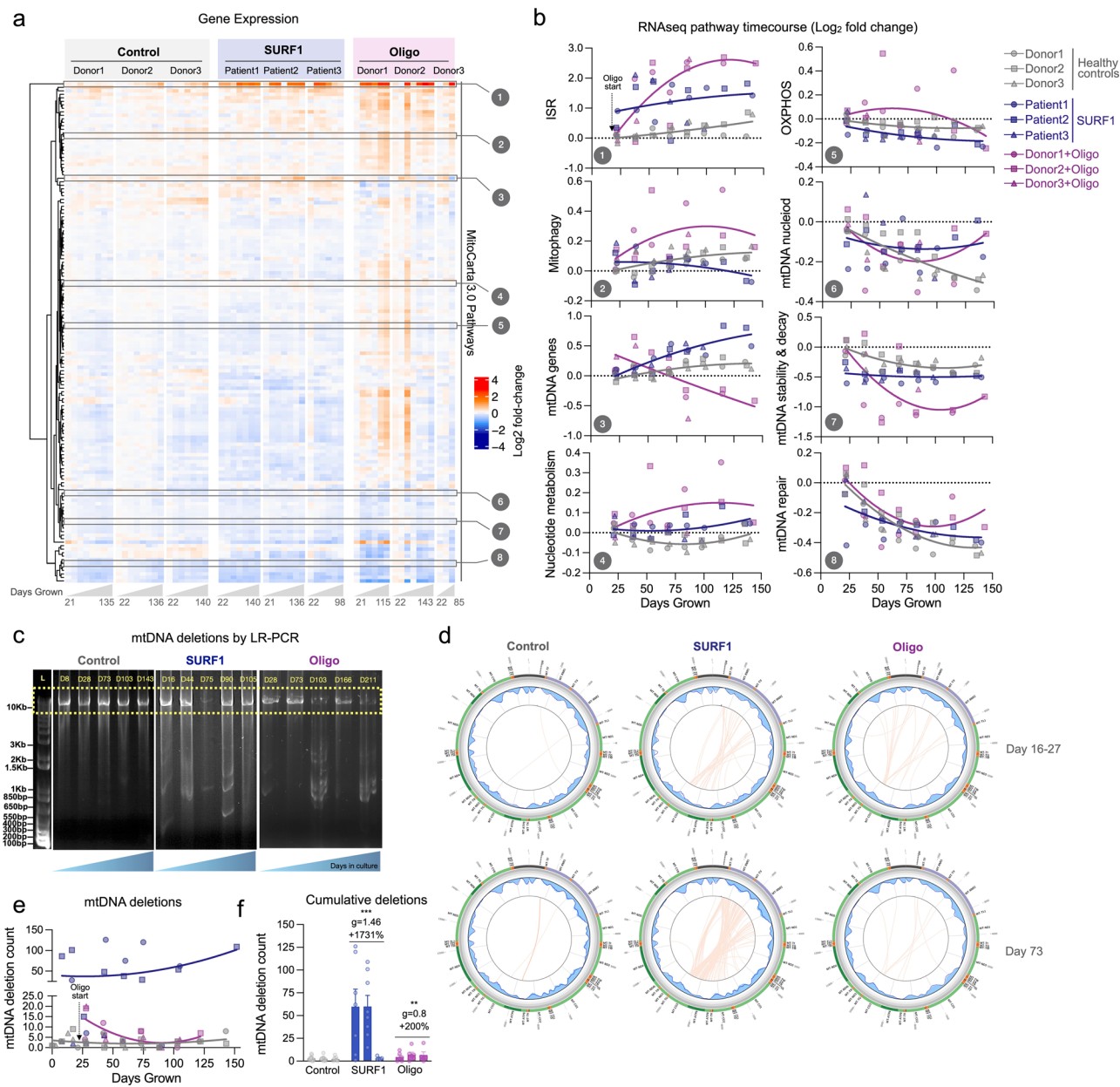

**Fig. 4 Longitudinal mtDNA deletion profiles in OxPhos deficient SURF1 and Oligo cells. a** RNAseq gene expression results for all MitoCarta 3.0 pathways, plus all mtDNA genes, and the integrated stress response (ISR: average of ATF4, ATF5, CHOP, GDF15). Values for each pathway are computed from the average expression levels of all genes in each pathway, expressed as the median-centered value relative to the youngest control timepoints for each pathway (rows). Each column represents a single timepoints ($n = 3$–8) along the lifespan of each donor or treatment condition ($n = 3$ donors, 3 groups). **b** Gene expression time course of selected mitochondrial pathways from E, expressed on a Log2 scale relative to the first control timepoint (baseline). **c** 10 kb long-range PCR product resolved by agarose gel electrophoresis for control fibroblasts cultured up to 166 days (P3 to 31), and passage-matched SURF1 and Oligo-treated cells. **d** Results from mtDNA sequencing and Eklipse analysis. Each line in the circos plots depicts a deletion burden in control (Donor2) and SURF1 (Patient2) and Oligo-treated (Donor2) cells from two (early and mid-lifespan) representative passages. **e** Timecourse of the number of unique mtDNA deletions in control, SURF1, and Oligo-treated cells. Deletion counts were estimated with a variant call cutoff of >5% heteroplasmy. **f** Total deletion burden in cells across 150 days of lifespan. Data are mean ± SEM. **$P < 0.01$, ***$P < 0.001$, mixed effects model (fixed effect of Control/SURF1/Oligo group and days grown, random effects of donor or cell line).

**OxPhos defects upregulate energy-demanding cellular programs.** From our longitudinal RNAseq dataset, we noted changes in the totality of genes related to the ribosomal machinery, which is produced in proportion with cellular biosynthetic demands (Supplementary Figure 10a). Despite their significantly reduced growth rate, both SURF1 (+19%, $p = 2.4e − 10$, $g = 2.59$) and Oligo-treated cells (+50%, $p < 0.01$, $g = 0.86$) showed a marked time-dependent upregulation in the ribosomal machinery, consistent with the hypersecretory phenotype (secreted proteins must

be transcribed and translated) as well as the elevated metabolic demands of translation that compete with cell growth[44,76] (Supplementary Fig. 10b). To characterize the genome-wide gene regulatory changes associated with these cellular phenotypes and to gain insights into the potential cause(s) of hypermetabolism in response to OxPhos defects, we next deployed time-sensitive models of gene regulation and DNA methylation.

We first visualized the transcriptomic profiles of SURF1 and Oligo-treated cells using t-distributed stochastic neighbor

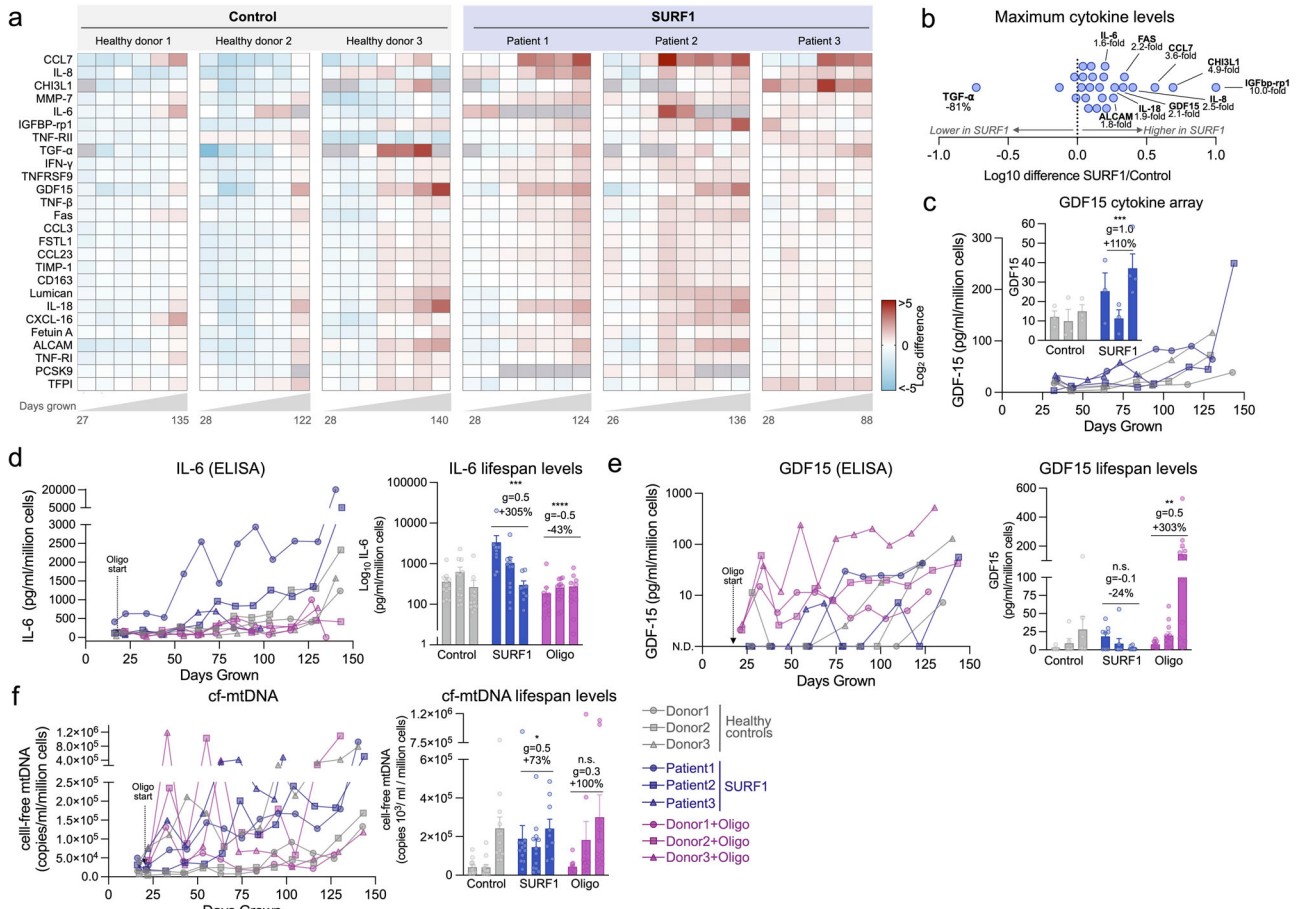

**Fig. 5 OxPhos defects trigger hypersecretion of metabokines and age-related cytokines. a** Cytokine dynamics across the lifespan measured on two multiplexes (Luminex) arrays. Cytokine levels are normalized to the number of cells at the time of sampling, shown as Log$_2$ median-centered for each cytokine; samples with undetectable values are shown as gray cells. Columns represent repeated measures (n = 6–8) along the lifespan of each control and SURF1 donor (n = 3 per group). **b** Comparison of maximum cytokine concentration reached in each of the SURF1 and healthy control donors, showing general upregulation of most metabokines and cytokines. The value for TGF-α is heavily influenced by a single very high value in Donor 3. **c** Cell-free media GDF15 concentration time course as measured on the Cytokine array. Inset compares early release between 20 and 80 days. **d** Media IL-6 levels across the cellular lifespan by enzyme-linked immunosorbent assay (ELISA), normalized to the number of cells at the time of sampling. **e** Media GDF15 levels across the cellular lifespan measured by ELISA, normalized to the number of cells at the time of sampling. Samples with non-detectable values (N.D.) are shown as zero values. **f** Cell-free mitochondrial DNA (cf-mtDNA) dynamics across the cellular lifespan using qPCR, normalized to the number of cells at the time of sampling. n = 3 per group, 6–13 timepoints per condition. Data are means ± SEM. *P < 0.05, **P < 0.01, ***P < 0.001, ****P < 0.0001, mixed effects model (fixed effect of Control/SURF1/Oligo group and days grown, random effects of donor or cell line). Abbreviations: *CCL7* C-C motif chemokine ligand 7, *IL-8* interleukin 8, *CHI3L1* chitinase-3-like protein 1, *MMP7* Matrix metallopeptidase 7, *IL-6* Interleukin 6, *IGFBP-rp1* Insulin-like growth factor binding protein 7, *TNF-RII* tumor necrosis factor receptor superfamily member 1B, *TGF-α* tumor growth factor alpha, *IFN-γ* interferon-gamma, *TNFRSF9* TNF receptor superfamily member 9, *GDF-15* growth differentiation factor 15, *TNF-β* tumor necrosis factor beta, *Fas* Fas cell surface death receptor, CCL3 C-C motif chemokine ligand 7, FSTL1 Follistatin like 1, CCL23 C-C motif chemokine ligand 23, TIMP-1 tissue inhibitor of metallopeptidase 1, *CD163* CD163 antigen, *Lumican* keratan sulfate proteoglycan Lumican, *IL-18* interleukin-18, *CXCL16* C-X-C motif chemokine ligand 16, *Fetuin A* alpha 2-HS glycoprotein, *ALCAM* activated leukocyte cell adhesion molecule, *TNF-RI* TNF receptor superfamily member 1A, *PCSK9* proprotein convertase subtilisin/kexin type 9, *TFPI* tissue factor pathway inhibitor.

embedding (t-SNE). Spatial embedding along the two major tSNE components captured three main features of the transcriptome: (i) substantial interindividual differences separating each donor/cell line, (ii) age-dependent shifts in transcriptional profiles, (iii) clustering among both SURF1 and Oligo cells, consistent with a main effect of OxPhos defects (Fig. 6a). To harness the longitudinal nature of these data, we used a linear mixed effects model (LMER) to identify time-dependent differentially expressed genes (DEGs, FDR < 0.05 threshold) between SURF1 and Oligo relative to control, across the cellular lifespan (Supplementary Data 1 and 2). Consistent with the similar degree of hypermetabolism and the metabolic shift among both cellular models (see Figs. 2d–f and 3b–f), there was a relatively high degree of overlap in DEGs between SURF1 and Oligo-treated cells (Supplementary Figs. 11

and 14 and Supplementary Data 3). Genes with the largest effect sizes conserved across SURF1 and Oligo showed up to 2-4-fold upregulation (39%, n = 1503) or downregulation (35%, n = 1344) (Fig. 6b, c). Differences were larger and more stable between SURF1 and control, compared to more progressive effects following the beginning of the Oligo treatment (Supplementary Fig. 11), consistent with the constitutive genetic deficiency in SURF1 cells compared to the novel insult with Oligo treatment. The influence of OxPhos defects on the expression of the 37 mtDNA genes across the cellular lifespan is shown in Supplementary Fig. 12.

To identify gene regulatory pathways associated with hyper-metabolism, we analyzed gene expression changes using iPAGE, an information-theoretic computational framework that enables

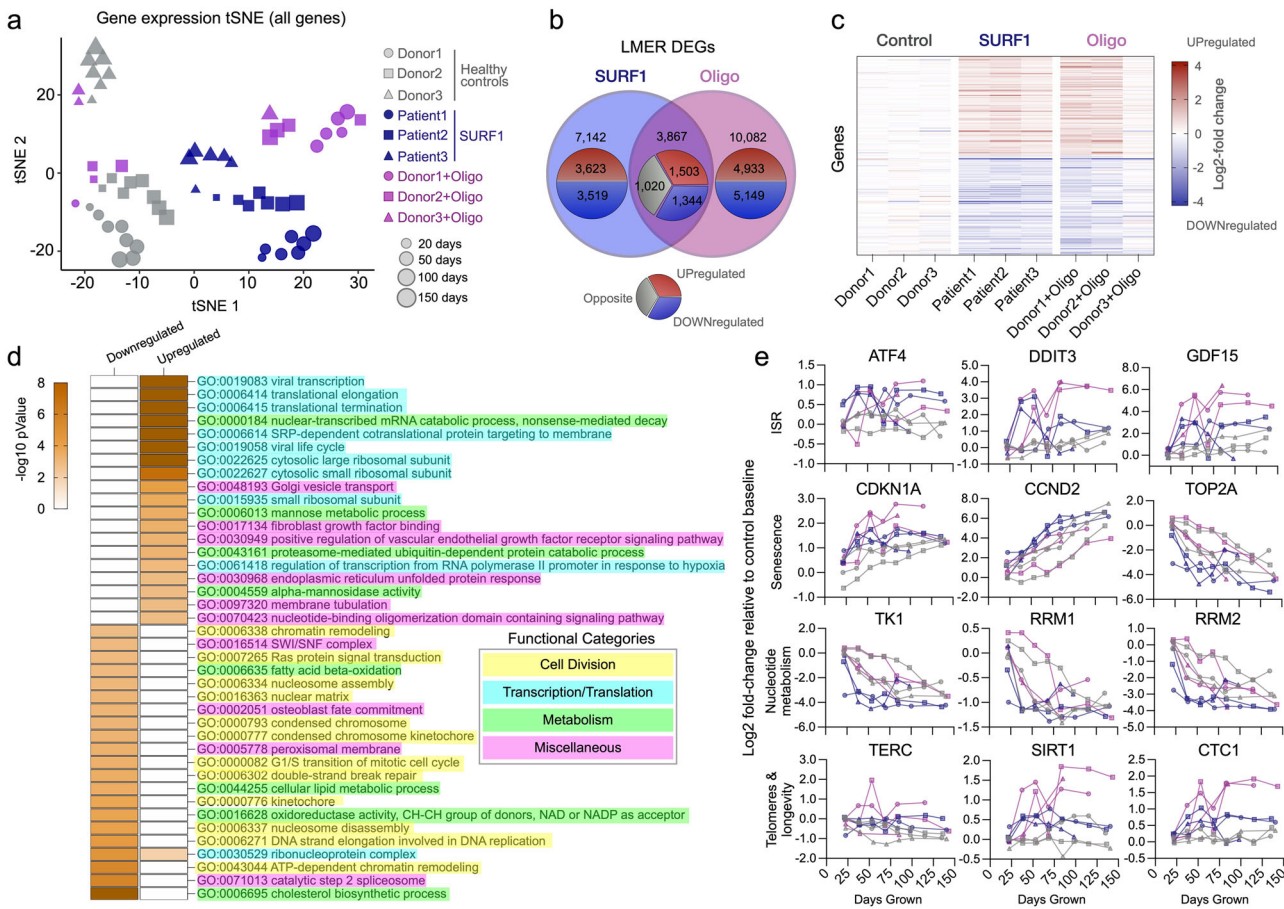

**Fig. 6 Mitochondrial defects trigger conserved transcriptional remodeling. a** t-distributed stochastic neighbor embedding (t-SNE) of RNAseq data from control, SURF1, and Oligo-treated human fibroblasts across the lifespan. **b** Overlap of significantly upregulated (red) or downregulated (blue) genes in SURF1 and Oligo groups relative to control (linear mixed effects model, FDR-corrected $p$ value < 0.05). Note, outer group counts include shared counts in the overlapping rings. Gray indicates the diverging direction of regulation between SURF1 and Oligo DEGs. **c** Expression levels of the top 100 differentially expressed genes in SURF1 (<75 days grown) and Oligo-treated cells (days 35–110). **d** iPAGE analysis of RNAseq data showing the top 40 enriched gene ontology pathways in top overlapping upregulated and downregulated genes, conserved across both SURF1 and Oligo groups relative to control. Note, −log($p$ value) > 8 are mapped as dark orange. **e** Gene expression timecourses of select genes related to the ISR, senescence, nucleotide metabolism, and telomere maintenance. Log2 expression values (TPM) are normalized to the median of the control youngest timepoints. $n = 3$ donors per group, 3–8 timepoints per donor.

the systematic discovery of perturbed cellular pathways from gene expression data[77]. Both SURF1 and Oligo-treated cells displayed a significant perturbation of transcription and translation processes (Fig. 6d). Upregulated genes were enriched for pathways related to Golgi vesicle transport, fibroblast growth factor (FGF) binding, *VEGF* receptor signaling pathway, and the unfolded protein response, a signature consistent with increased secretion and inter-cellular signaling activity. Downregulated genes were overrepresented for processes relating to cell division, consistent with the slower division rates (i.e., quiescence or senescence) of SURF1 and Oligo-treated cells.

Lifespan gene expression trajectories in this dataset showed some noteworthy features of OxPhos defects and hypermetabolism at the single-gene level: (i) ISR-related genes are robustly upregulated in a time-dependent manner by up to ~16-fold for the transcription factor *CHOP* (*DDIT3*), and ~60-fold for its downstream target *GDF15*; (ii) the age-related upregulation of senescence-related genes (e.g., *p21/CDKN1A*) occurs prematurely in hypermetabolic SURF1 and Oligo cells; (iii) key nucleotide metabolism enzymes such as thymidine kinase 1 (*TK1*) are robustly downregulated in SURF1 cells, possibly contributing to mtDNA instability[78]; and (iv) telomere and longevity-related genes such as the metabolic sensor SIRT1 and

the telomere protection complex component *CTC1* are upregulated two to fourfold (Fig. 6e). These broad changes in gene expression, largely consistent with previous in vitro work[3,79], prompted us to examine another major layer of gene regulation, DNA methylation.

**DNA methylation recalibrations in OxPhos-induced hypermetabolism.** To examine nuclear DNA methylation (DNAm) and create a resource dataset with broad utility for pathway discovery, we measured DNA methylation levels at 865,817 CpG sites (Illumina EPIC array) in Control, SURF1, and Oligo-treated cells at multiple time points across their cellular lifespan ($n = 66$). We then leveraged these high dimensional data by building mixed-effects models that consider the underlying data structure (donors, longitudinal observations) to identify robust conserved DNA methylation changes associated with OxPhos defects and hypermetabolism. Visualizing the general data structure using t-SNE showed, that: (i) as expected, the methylome signature of each donor was relatively distinct; (ii) DNAm exhibited consistent age-related shifts, (iii) SURF1 cells clustered separately from control, while (iv) Oligo cells caused a modest time-dependent shift away from their respective controls (Fig. 7a). These data, therefore, add

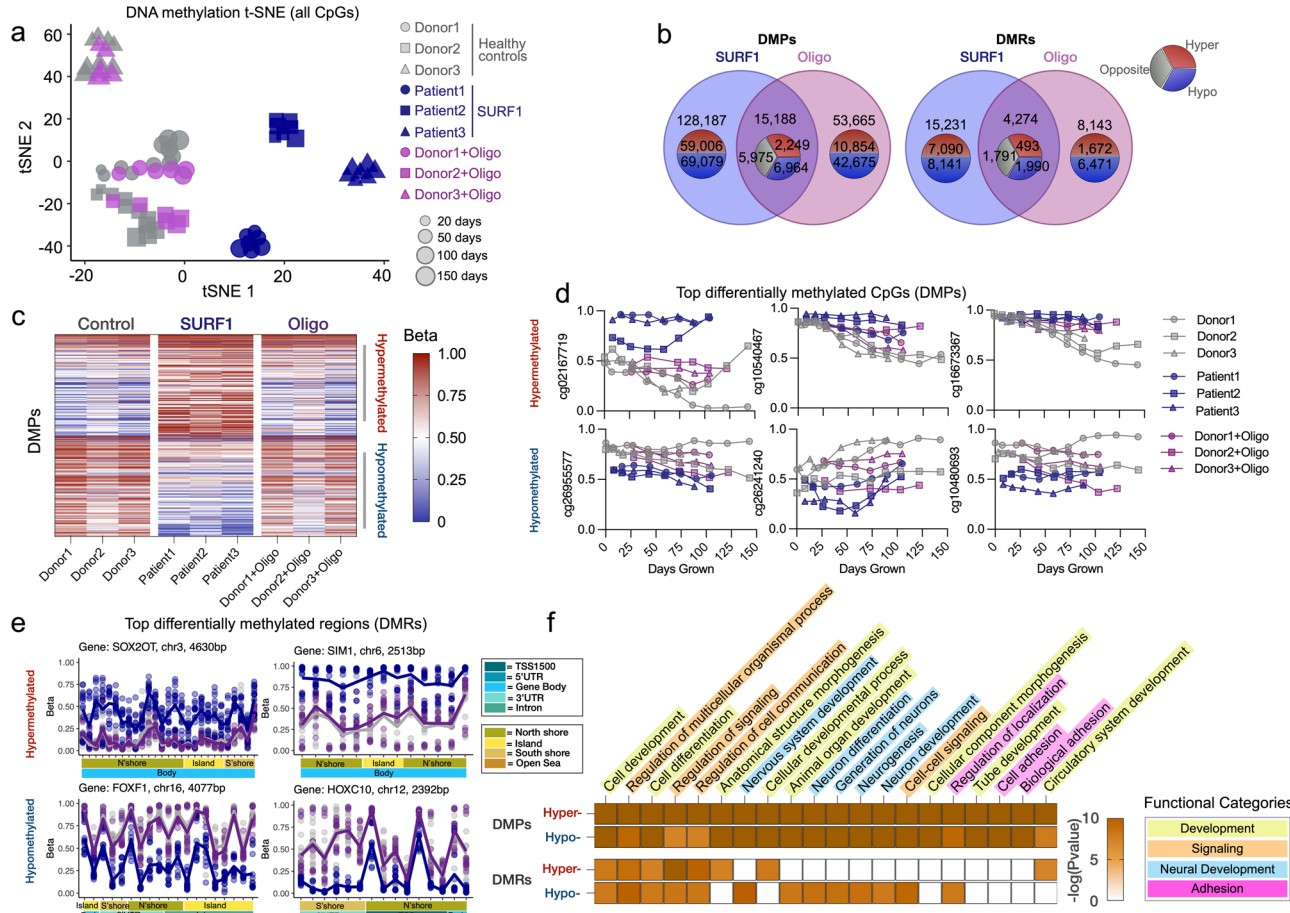

**Fig. 7 Mitochondrial defects trigger conserved epigenetic remodeling. a** t-SNE of the nuclear DNA methylome from control, SURF1-disease (<75 days grown), and Oligomycin-treated (35–110 days grown) fibroblasts across the replicative lifespan. **b** Overlap of differentially methylated CpGs (DMPs, left panel) and differentially methylated regions (DMRs, right-panel) generated from mixed effects models. Note, outer group counts include shared counts in the overlapping rings. **c** Heatmap of top 100 DMPs in SURF1 and Oligo-treated cells. DMPs were ordered by mean methylation difference between groups. **d** Timecourse of top three hyper- and hypo-DMPs for SURF1 and Oligo-treated groups. **e** Gene regional map of top hyper- and hypo-DMRs for SURF1-disease and Oligo-treated fibroblasts. 5'–>3' direction. **f** Heatmap of top 20 enriched gene ontology pathways in top 1000 hyper- and hypo-DMPs and DMRs overlapping between SURF1 and Oligo-treated groups. Note, −log(P values) > 10 are mapped as dark orange. n = 3 donors per group, 5–11 timepoints per donor/treatment.

to previous evidence in HEK293 cells[80] and mice[81], providing a robust platform for discovering conserved nuclear DNAm signatures associated with hypermetabolism-causing OxPhos defects in primary human cells.

At the single CpG level, we asked which differentially methylated positions (DMPs) were stably and consistently either hypo- or hypermethylated in both SURF1 or Oligo-treated cells relative to the control. Because transcriptionally relevant DNAm changes may operate across multiple CpGs, we complemented this approach by systematically examining differentially methylated regions (DMRs), which include multiple nearby CpGs exhibiting similar hypo- or hypermethylated changes in our statistical model[82] (see Methods for details). Figure 7b shows the overlap in significant DMPs and DMRs (threshold FDR < 0.05). Of the overlapping DMPs between SURF1 and Oligo, 14.8% were hypermethylated, and 45.9% were hypomethylated. Global hypomethylation is a feature of human aging and replicative senescence[57]. For DMRs, the corresponding proportions were 11.1% and 46.6%, showing high agreement in the methylome recalibrations between DMPs and DMRs approaches. A notable number of significant and highly differentially methylated changes in either SURF1 or Oligo-treated cells were specific to each condition (Supplementary Data 4–9), but here we focus exclusively on the changes conserved across two

independent models, which therefore have the highest probability of being specifically caused by OxPhos defects and associated with hypermetabolism (Fig. 7c, d).

The most robust changes in DNA methylation were targeted at CpG islands near or on gene bodies. Relative to control cells, as in the RNAseq results, the effect sizes were larger for SURF1 compared to Oligo, which induced directionally consistent but smaller effect size changes than *SURF1* defects (Fig. 7e). A stringent analysis of the most differentially methylated genes (based on both DMPs and DMRs) showed strong enrichment for processes involving: (i) development and morphogenesis, (ii) regulation of cell-cell signaling and organismal communication, (iii) neural development, and (iv) cell adhesion (Fig. 7f). As highlighted above, increased regulation of signaling and communication, along with development and morphogenesis, must entail energetically dependent processes. These data, supported by the activation of corresponding downstream transcriptional programs (Fig. 6) and the observed hypersecretory phenotype in OxPhos-deficient cells (Fig. 5), document genome-wide epigenomic recalibrations consistent *not* with energy conservation, but with increased total energy expenditure. These data also can be further queried with specific genomic targets in mind.

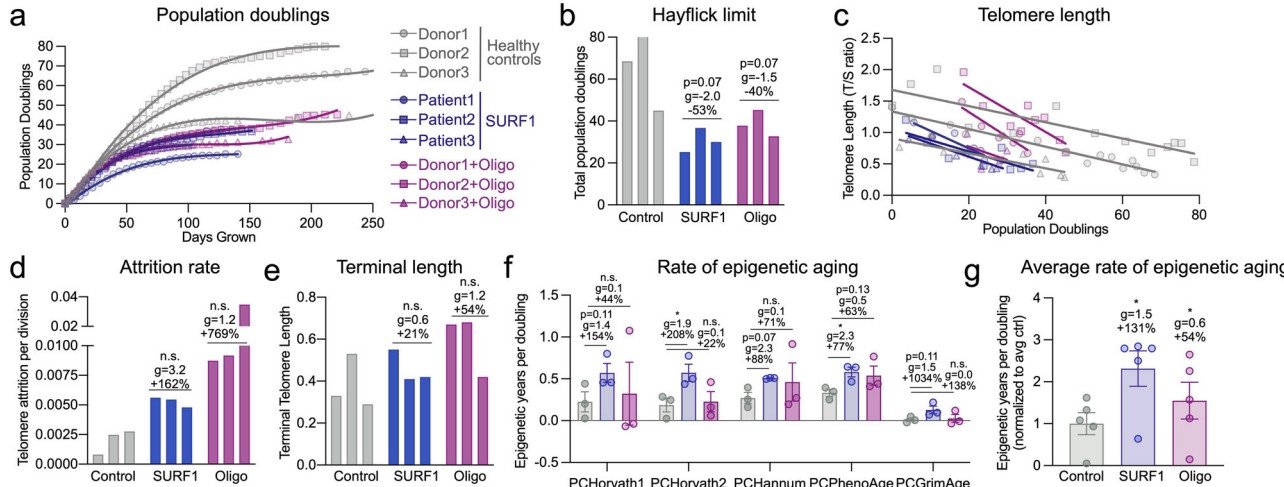

**Fig. 8 Mitochondrial OxPhos defects decrease lifespan and accelerate telomere shortening. a** Growth curves of control, SURF1, and Oligo-treated cells. Population doublings were determined from both live and dead cell cells at each passage. **b** Hayflick limit defined as the total number of population doublings achieved before division rate <0.01 divisions/day for at least two passages. **c** Telomere length per population doubling, **d** rate of telomere attrition per division, and **e** terminal telomere length. **f** Rate of epigenetic aging for control, SURF1, and Oligo-treated cells, calculated from the linear rate between days 25 and 75 (3–4 timepoints/cell line). **g** Average rate of epigenetic aging across all PC-based clocks. Each datapoint represents a different clock. **f**, **g** Significance values were calculated using a multiple comparison two-way ANOVA. $n = 3$ donors per group, 5–15 timepoints per condition for telomere length. In **d**, data are the slope estimate for the linear regressions in (**c**). Data are means ± SEM. *$P < 0.05$, **$P < 0.01$.

## OxPhos defects accelerate telomere shortening and decrease cellular lifespan

Finally, given the deleterious effect of hypermetabolism-causing OxPhos defects on the lifespan of patients with mitochondrial diseases and in animal models, these genome-wide data prompted us to examine how OxPhos defects and hypermetabolism relate to dynamic genomic markers of cellular aging and senescence. The complete population doubling curves of each donor (Fig. 8a) provided initial evidence that cellular lifespan was reduced in SURF1 and Oligo-treated cells. The Hayflick limit (i.e., the total number of cell divisions[56]) was, on average, 53% lower in SURF1 cells ($p = 0.072$, $g = 2.0$), and Oligo decreased the Hayflick limit by 40% ($p < 0.066$, $g = 2.0$) relative to the untreated cells of the same donor (Fig. 8a, b). Interestingly, the magnitude of these effects (40–53%) on total population doubling loosely corresponds to the 3–4 decade loss in human lifespan documented among adults with mitochondrial diseases (see Fig. 1g, h), which would represent 38–50% for an average 80-year life expectancy.

To directly measure the pace of biological aging in response to OxPhos defects, we performed repeated measures of telomere length across the cellular lifespan. This allowed us to compute the average *rate* (i.e., slope) of telomere shortening per population doubling or cell division (Fig. 8c). Consistent with observations of dramatically shortened telomeres in skeletal muscle of patients with mtDNA mutations[83] and recent work causally linking mitochondrial OxPhos defects to telomere dysfunction[84], both SURF1 mutations and Oligo treatment strikingly increased the rate of telomere erosion per population doubling by 162% for SURF1 ($p = 0.53$, $g = 3.2$) and 769% for Oligo ($p = 0.09$, $g = 1.2$) (Fig. 8d). This means that for each cell division and genome duplication event, OxPhos-deficient fibroblasts lose 1.6–7.7 times more telomeric repeats than healthy fibroblasts. We note that these results rely on the estimated slope across the whole cellular lifespan (single value per donor, $n = 3$ per group), so the $p$ values are less meaningful than the effect sizes, which are large ($g > 1$). The terminal telomere length coinciding with growth arrest tended to be moderately higher in SURF1 and Oligo groups (Fig. 8e). This could suggest that growth arrest is driven by factors other than absolute telomere lengths, such as the prioritization of transcription/translation over growth-related functions, which are sufficient to induce growth arrest and senescence in human fibroblasts[44,45].

Next, we leveraged our DNAm dataset to quantify biological age using validated multivariate algorithms or "clocks" (DNA-mAge, or epigenetic clocks) trained, in human tissues, to predict chronological age and mortality[57,85]. Five different DNAmAge clocks that rely on different CpG sets and include a modification that improves their accuracy[86] were applied directly to our fibroblast time series DNAm data. These results showed that relative to the rate of epigenetic aging in control cells with normal OxPhos function, the rates of biological aging per population doubling were accelerated by an average of 131% in SURF1 cells ($p < 0.05$, $g = 1.5$), and to a lesser extent in Oligo-treated cells (+54%, $p < 0.05$, $g = 0.6$, Fig. 8f, g), thus independently supporting the findings of accelerated telomere shortening. Trajectories and DNAm aging rates for each donor using all five epigenetic clocks, including those computed relative to "time in culture" rather than to population doublings, produced variable results and are presented in Supplementary Fig. 13.

Together, the decreased Hayflick limit, the accelerated telomere attrition rate, and the increased rate of epigenetic aging converge with the senescence-related secretome and gene expression results to link OxPhos defects to hypermetabolism and reduced cellular lifespan.

## Discussion

Integrating available clinical and animal data together with our longitudinal fibroblast studies has revealed hypermetabolism as a conserved feature of mitochondrial OxPhos defects. A major advantage of our cellular system is that it isolates the stable influence of genetic and pharmacological OxPhos perturbations on energy expenditure, independent of other factors that may operate in vivo. Thus, these data establish the cell-autonomous nature of hypermetabolism. Moreover, despite the diverging mode of action of SURF1 and Oligo models, as well as some divergent molecular responses, both models converge on the same hypermetabolic phenotype, adding confidence around the

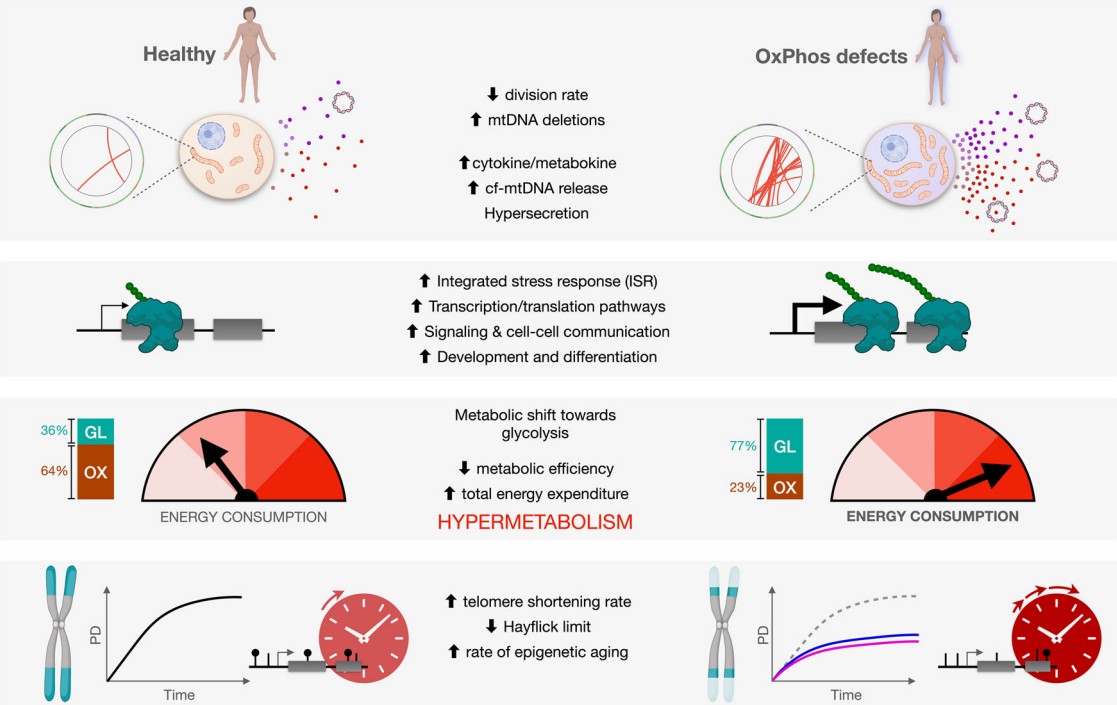

**Fig. 9 Conceptual model including potential sources of hypermetabolism in cells and patients with mitochondrial diseases.** OxPhos defects trigger mtDNA instability and cell-autonomous stress responses associated with the hypersecretory phenotype, recapitulating findings in plasma of patients with elevated metabokine and cell-free mitochondrial DNA (cf-mtDNA) levels. These responses are linked to the upregulation of multiple energy-dependent transcriptional programs, including the integrated stress response (ISR). We propose that these processes collectively increase energy consumption, leading to *hypermetabolism* in patient-derived fibroblasts, and physiological hypermetabolism in affected patients. In dividing human fibroblasts, hypermetabolism-causing OxPhos defects curtail lifespan and accelerate canonical cellular senescence and aging markers, namely telomere length, epigenetic aging, as well as secreted and transcriptional markers.

generalizability of this phenomenon. Our data also rule out mitochondrial uncoupling as a main driver of hypermetabolism in SURF1 patient-derived fibroblasts, and instead implicate the activation of energy-demanding gene regulatory programs, including but likely not limited to increased metabokine/cytokine secretion, which can compete with growth and longevity (Fig. 9). Our resource cellular lifespan data provide several novel observations that agree with previous work[79], and that are relevant to understanding how primary mitochondrial OxPhos defects triggers core physiological and phenotypic hallmarks of aging and mitochondrial diseases.

First, we observed that the mitochondrial disease marker GDF15 was largely undetectable in the media of young, healthy fibroblasts, but increased progressively across the cellular lifespan. This finding recapitulates the age-related increase in GDF15 in humans[69,73] and adds to previous evidence of conserved age-related changes in DNA methylation in primary human fibroblasts cultured over several months[57]. Consistent with the higher GDF15 levels in primary OxPhos disorders in humans[72] and mice[10], extracellular GDF15 tended to be elevated in both models of OxPhos defects. Likewise, OxPhos defects increased extracellular cf-mtDNA levels, in line with recent reports that cf-mtDNA is elevated in primary OxPhos disorders[75] and with aging in humans[74]. The link between OxPhos defects and cf-mtDNA release requires further investigation.

Second, we observed that OxPhos dysfunction from *SURF1* mutations, and to a lesser extent Oligo treatment, both caused secondary mtDNA instability. mtDNA instability was associated with the variable accumulation of mtDNA deletions, but not point mutations, across the cellular lifespan. Our confidence in this result is reinforced by the longitudinal nature of the mtDNA

sequencing data, from the same primary cell lines examined at multiple time points. Notably, the time course data also showed that cell populations can eliminate a large fraction of mtDNA deletions within 12–14 days (mtDNA deletions are removed from one passage to the next). This is consistent with the fact that replicating fibroblasts eliminate some deleterious mtDNA deletions[87], and also that several *de novo* deletions removed the origin of replication of the light strand ($O_L$), thereby preventing their replication. Whether the clonal amplification of some mtDNA deletions in SURF1 fibroblasts occurs through population selection at the cellular level, through intracellular quality control mechanisms, or a combination of both, remains to be determined.

Third, mitochondrial OxPhos defects dramatically increased the telomere erosion rate per cell division, despite the adaptive transcriptional upregulation of telomere protection complex components. This effect of mitochondria on telomeres agrees with the variable telomere maintenance in mtDNA conplastic mice[88], with the life-shortening effect of pathogenic mtDNA variants[32] and OxPhos defects in mice[34], and with the reduced lifespan in patients with mtDNA disease shown in Fig. 1g, h. A study in skeletal muscle of children with high heteroplasmic mtDNA mutations also reported excessively short telomeres, similar in length to the telomeres of healthy 80-year-old controls[83]. Because skeletal muscle is a post-mitotic tissue, this previous result also implies that OxPhos defects could accelerate telomere attrition at a disproportionate rate, or perhaps independent from cell division, as suggested by the disconnect between the loss of telomeric repeats and genome replication/cell division observed in our hypermetabolic fibroblasts. Beyond severe OxPhos defects, mild alterations of OxPhos function

driven by mild, common variants in complex I subunits genes, may also shape disease risk[89] and influence lifespan[90].

The mechanistic link between OxPhos-induced hypermetabolism and both mtDNA instability and accelerated telomere erosion remains unclear. DNA maintenance (mtDNA, and telomeres) relies on the accuracy of the molecular processes ensuring accurate replication. The energetic tradeoff or "competition" between translation and growth[44] could explain why OxPhos-deficient cells, which must expend a large fraction of their energy budget to upregulate transcription/translation and secretory, also grow more slowly. DNA replication is also energetically constrained and notably sits at the bottom of a hierarchy of energy-consuming processes where vital processes, meaning that in a situation when energy is limited, ionic balance and translation are prioritized over division and DNA replication[20]. Furthermore, cells under stress experience an energetic tradeoff between the accuracy of molecular operations and the speed of these processes, known as the energy-speed-accuracy tradeoff[91]. Oxygen tension affecting electron flux through the OxPhos system could also contribute to imposing energetic tradeoffs. However, our cells grown at 3% oxygen did not show significantly different growth rates nor energy expenditure. On this basis, we can largely rule out a main effect of oxygen tension in our results and instead conclude that the hypermetabolic cellular phenotype is a direct consequence of the OxPhos defects. This interpretation is reinforced by the orthogonal nature of the SURF1 (complex IV) and Oligo (complex V) models, which target different OxPhos components yet produce a comparable degree of hypermetabolism. Hypermetabolism is a global *state* of the cell, and no currently available approach can selectively manipulate or correct hypermetabolism without introducing unresolvable confounds. For this reason, to our knowledge, it is currently not possible to mechanistically isolate the influence of hypermetabolism on DNA instability and accelerated telomere shortening. We speculate that the diversion of energetic resources[43], as well as substrates including nucleotides[78], may contribute to reduced DNA replication fidelity, which in turn could contribute to both mtDNA instability and telomere attrition, independent of cell division.

Fourth, our longitudinal RNASeq and DNAm datasets reveal conserved recalibrations implicating developmental and translation-related pathways, as well as cell–cell communication, with OxPhos defects and hypermetabolism. These identified pathways overlap with previously identified multi-omic over-representation analysis performed on iPSC-derived neurons from SURF1 patients[92]. In both this and our study, neural development, cell signaling, morphogenesis, cell cycle, and metabolism were the predominant processes altered in *SURF1*-related disease. The induction of these energetically-demanding pathways that constrain growth at the cellular and possibly at the organismal level[41], could help explain why a major feature of pediatric mitochondrial disorders (including our SURF1 donors) is a neurodevelopmental delay, and also why adult patients commonly display short stature (restricted growth)[30]. In relation to cell-cell communication, we note that the biomarker picture of adult patients with mitochondrial encephalopathy, lactic acidosis, and stroke-like episodes (MELAS) is dominated, as in our fibroblast models, by *elevated* (not reduced) signaling and metabolic markers in blood[72]. Thus, the organism under metabolic stress does not initiate an energy-saving *hypometabolic* state with reduced signaling activity, but instead activates energivorous stress responses (ISRs), which must divert and consume energetic resources, thereby forcing an apparent tradeoff with other processes such as growth and longevity pathways.

Finally, the OxPhos defects in our fibroblasts triggered a shift toward glycolytic ATP production. The glycolytic shift is consistent with the physiological shift in substrate oxidation from lipids/amino acids to carbohydrates, quantified by the RQ among patients[93] and mice[94] with OxPhos defects. The active shift towards glycolysis occurs even when OxPhos is *not* completely obliterated. For example, although basal respiration was markedly lower in SURF1 cells, the maximal FCCP-uncoupled respiration in SURF1 cells was relatively preserved (see Fig. 2b and Supplementary Fig. 2c). This result implies a cellular decision to route metabolic flux towards an energetically less efficient pathway (i.e., glycolysis). This could be explained on the basis of energetic constraints and proteome efficiency, since the proteome cost of OxPhos is at least double that of glycolytic fermentation[19]. Thus, cells can "choose" to divert metabolic flux towards glycolysis even when OxPhos is at least partially functional, as in cancer, because of rising intracellular energetic constraints driven by hypermetabolism. We note again that hypermetabolism is apparent across multiple animal models of primary OxPhos defects, manifesting as an elevated cost of living, even during rest and sleep in mice[10,24–26]. In particular, deep phenotyping of $Ant1^{-/-}$ mice across three studies[25,95,96] reveals a systemic physiological picture highly consistent with mitochondrial diseases, including excessive mitochondrial biogenesis, elevated circulating catecholamine levels, severe hypermetabolism (+82 to −85% REE) when adjusted for lower physical activity levels, reduced adiposity, elevated mtDNAcn, and mtDNA instability, and decreased median lifespan. These in vivo data thus provide additional converging evidence, beyond the clinical data in Fig. 1, that mitochondrial OxPhos defects impair whole-body energetic efficiency and cause physiological hypermetabolism in mammals.

Identifying hypermetabolism as a feature of the mitochondrial diseases may be clinically relevant as it provides an explanatory framework for some of the major symptoms in affected patients.

*First*, fatigue and exercise intolerance are evolutionary conserved, subjective experiences that arise when the organism consumes more energy than it would under optimal conditions (e.g., subjective fatigue during the oxygen debt after strenuous exercise, or during an infection). Thus, symptoms of fatigue could be direct consequences of impaired metabolic efficiency and hypermetabolism.

*Second*, as noted above, severely affected patients with mitochondrial disease are usually thin, which may be attributable to not only reduced energy intake or to intestinal malabsorption, but to chronic hypermetabolism, effectively burning excess ingested calories, preventing the accumulation of excess adiposity and muscle mass.

*Third*, alcohol appears to be poorly tolerated and associated with symptom onset in some patients with mtDNA defects[97–99], but the basis for alcohol intolerance remains unknown. Alcohol itself causes hypermetabolism in healthy individuals—increasing whole-body REE by as much as 16%, and inhibiting lipid oxidation by 31–36%[100,101]. Alcohol may therefore aggravate pre-existing hypermetabolism, thus imposing further energetic constraints on vital cellular or physiological functions.

*Fourth*, chronic hypermetabolism could, in part, explain why infections can trigger clinical exacerbations, representing the major cause of decompensation and death in this population[29]. The metabolic cost of immune activation to viral and bacterial infection is high, and cytokine production in human leukocytes is under mitochondrial regulation[102]. Thus, immunity must therefore compete with other host maintenance systems[103]. We speculate that in mitochondrial diseases, because the limited energetic resources are consumed at a higher rate than normal due to systemic hypermetabolism, patients may lack the necessary energetic reserve required to sustain vital organs while mounting adequate immune responses.

A major open question relates to the origin and modifiability of signaling pathway(s) and cellular process(es) that underlie

hypermetabolism in OxPhos deficient cells and humans. Rather than pursuing a single potential explanation, here we attempted to deeply phenotype both cellular models of hypermetabolism and to produce a foundational dataset covering several key processes and pathways previously implicated in the pathogenesis of OxPhos defects in humans and animal models. Our dataset, therefore, provides a foundation that can be used as a resource to develop targeted, mechanistic experiments to (i) determine the origin and modifiability of hypermetabolism in the context of OxPhos defects in vitro and in vivo, and (ii) resolve the mechanism(s) linking hypermetabolism to human aging biology. A usual suspect to explain decreased cellular and physiological bioenergetic efficiency is the uncoupling of the OxPhos machinery. While this article was under revision, a case of clinically suspected hypermetabolism in twin boys harboring a complex V ($F_OF_1$ ATP synthase) defect was reported[104]. Introducing the pathogenic mutation in fibroblasts increased basal OCR, where the underlying mechanism involves elevated proton leak and uncoupling. This agrees with the first reported case of mitochondrial disease by Luft, a woman with uncoupled skeletal muscle mitochondria suffering from severe, documented hypermetabolism[105]. This uncoupling phenotype differs from our observations, which indicate no sign of uncoupling in SURF1-mutant hypermetabolic fibroblasts – in fact, significantly lower (−35%) proton leak and higher (+10%) coupling efficiency (Fig. 2j, k). On the other hand, the Oligo treatment targeting the complex V did cause a marginally significant (−28%) reduction in coupling efficiency and punctual elevations in proton leak across the lifespan (Fig. 3g, h). These results point to a potential role of OxPhos uncoupling as a partial contributor to hypermetabolism in the Oligo model. However, other cellular data, as well as the physiological results from our meta-analysis where only a minority of patients had mutations affecting the $F_OF_1$ ATP synthase, call attention to uncoupling-independent, energy-demanding processes as more general causes of hypermetabolism deserving further investigation.

Factors that could be regarded as limitations of this study include the small sample size and the in vitro nature of the cellular data, potentially limiting generalizability. However, the stability of metabolic and molecular phenotypes in two distinct experimental models, across three unrelated donors (female and male) systematically monitored across the replicative lifespan—when cells undergo dynamic age-related changes—is a strong test of robustness for these findings. We also note that the extracellular flux analysis used to derive ATP consumption rates is indirect[60], and other approaches, such as metabolic tracing experiments, would be required to fully define energy partitioning in hypermetabolic cells. Finally, the clinical phenotyping presented in Fig. 1 is not exhaustive, focusing exclusively on available clinical outcomes related to energy expenditure, including indirect calorimetry without careful body composition or age adjustments. Further studies are therefore needed to address potential confounders and to fully define the magnitude and clinical heterogeneity in energy expenditure among patients, together with the dynamic neuroendocrine and metabolic manifestations of hypermetabolism.

Overall, the meta-analysis of clinical data from hundreds of patients and two cellular models of OxPhos dysfunction identifies hypermetabolism as a feature of mitochondrial diseases. Our longitudinal patient-derived fibroblasts data delineate some of the cellular and molecular features of OxPhos-induced hypermetabolism, including sustained induction of the ISR, genome instability, hypersecretion of cyto/metabokines, and genome-wide DNA methylation and transcriptional recalibrations that emphasize the upregulation of energy-dependent processes related to signaling and communication (see Fig. 9). A resource webtool with all data from this study, including the longitudinal RNAseq and DNAm data, is available and can be explored for genes or processes of interest (see *Data Availability Statement*). Altogether, these translational data, therefore, provide a basis to rationalize some unexplained clinical features of mitochondrial diseases and suggest that intracellular and systemic energy tradeoffs (rather than ATP deficiency) may contribute to the pathogenesis of mitochondrial diseases. The proposed explanatory framework of cellular and physiological hypermetabolism calls for well-controlled studies to further understand the extent to which hypermetabolism is a bystander or a harbinger of morbidity and early mortality in patients with mitochondrial diseases. Our translational findings highlight the need for collaborative partnerships that bridge the cellular, clinical, and patient-reported aspects of mitochondrial diseases and aging.

## Methods

**Human cohorts**. Data were meta-analyzed from 17 mitochondrial disease cohorts listed in Table 1. Inclusion criteria included (1) cohorts with a genetic diagnosis for all participants and (2) including measures for at least one of the primary outcomes (resting heart rate, catecholamine levels, resting $VO_2$ or $VO_2$ relative to work rate, BMI, mortality). Eligible cohorts included participants from five countries, including China, Denmark, England, Italy, and the USA. Studies were published between 2003 and 2019, covering a 16-year period. Each cohort with its sample size, female/male distribution, genetic diagnoses (nDNA vs mtDNA), and symptomatology are listed in Table 1, with additional information about data extraction provided here. Each cohort included its own control group, so group-level averages (not patient-level data) were used to compute effect sizes as % difference between mitochondrial diseases and control, and standardized Hedges g for each outcome measure (e.g., resting heart rate, resting $VO_2$). Cohorts with available source data to calculate intragroup variance include error bars denoting the standard error of the mean in Fig. 1.

Cohort 1[54] included data on resting HR and resting $VO_2$ in patients with mixed genetic defects. Cohort 2 included four sub-studies: (a)[106], (b)[107], (c)[108], and (d) a cohort of patients with single large-scale mtDNA deletions with measures of resting HR, resting $VO_2$, and BMI. Cohort 3[109] included data on resting HR, resting urinary catecholamines, and BMI. Cohort 4[110] included data on resting HR and BMI. Cohort 5 is a cohort (the Mitochondrial Stress, Brain Imaging, and Epigenetics Study – MiSBIE) of patients with m.3243 A > G mutations, which included data on resting HR and BMI. Cohort 6[111] included data on circulating catecholamines at rest and during exercise. Cohort 7[93] included data on $VO_2$ during fixed workload (65 W) and BMI. Cohort 8[112] included data on resting $VO_2$ and BMI. Cohort 9[113] included data on $VO_2$ during constant work rate (40% of max), and $VO_2$ values in ml/kg/min were adjusted to average workload achieved by each group to obtain comparable estimates of energetic demand relative to work performed. Cohort 10[14] included metabolic efficiency during constant-rate cycle ergometry (30 watts), including before and after a home-based exercise training protocol, and these values were compared to reference values in healthy individuals from[113]. Cohort 11[114] overlaps with Cohorts 9 and 10 and included BMI data. Cohort 12[17] included BMI data averaged between both mutation groups. Cohort 13[30] is a natural history study of adult patients with mortality data. Cohort 14[29] is a retrospective study of the causes of death in adult patients with mortality data. Cohort 15[115] is a pediatric natural history study with mortality data. Cohort 16[28] is a multi-center pediatric natural history study with mortality data. Cohort 17 is an ongoing natural history study (McFarland et al., Newcastle Mitochondrial Disease Cohort) with mortality data.

For a subset (3/6) of studies reporting both $VO_2$ and $VCO_2$ in the original publication, or reporting both $VO_2$ and the respiratory quotient (RQ) from which $VCO_2$ could be derived, we used the Weir equation[53] to estimate group-level REE differences between patients and controls. Compared to $VO_2$ (mlO$_2$/min/kg body mass) differences between groups, the Weir equation-derived REE differences (kcal/day/kg) were, on average, 1.2% higher (range: −0.3% to +2.3%) than the group difference in $VO_2$ (30.0%). Future studies using the proper methodology to quantify resting metabolic rate (RMR) or free-living energy expenditure, normalized with sensitive body composition assessments, are needed to fully define the spectrum of hypermetabolism in affected patients.

Reference BMI for the USA (29.9 kg/m$^2$) was obtained from the National Health and Nutrition Examination Survey (NHANES) for wave 2015-2016 ($n = 9544$) (e-link), for the UK (28.6 kg/m$^2$) from the Health Survey for England 2018 ($n = 6600$) (link), and for Italy (25.8 kg/m$^2$) from the NCD Risk factor collaboration (link), with the combined average presented in Fig. 1f. Reference values for life expectancy, were obtained from the World Bank (https://data.worldbank.org/) and the average value for the USA (78.6 yr), UK (81.2 yr), and Italy (82.9 yr) (representing most cohorts included) is reported in Fig. 1g. Data presented in Fig. 1h represent mortality rates in the UK (reference population) for 2018, and the mortality data for individuals for the mitochondrial disease was collected between 2010 and 2020.

The clinical data demonstrating hypermetabolism are derived from more than a dozen laboratories over a >15-year period, illustrating the stability of this finding. The apparent cross-study stability of clinical hypermetabolism is also unlikely to be influenced by publication or reporting bias for three main reasons: (i) most studies were exploratory (as opposed to confirmatory) in nature, such that the motivation for their publication depended neither on the significance nor direction of these results, (ii) baseline group differences for most parameters (e.g., resting $VO_2$) were not primary outcomes in any studies, and in several cases, these data were not analyzed nor reported in the original reports, and (iii) variables such as BMI were ubiquitously reported. Moreover, to further reduce the potential of bias, the overall sample includes new, previously unpublished cohorts of clinically and genetically well-defined patient populations (see Table 1). Together, these factors increase the likelihood that the findings revealing the existence of a hypermetabolic state are robust and generalizable to mitochondrial diseases represented here, which includes a relatively broad diversity of mtDNA mutations. Further work is needed to sensitively quantify hypermetabolism across the diurnal cycle, normalized to body composition (fat-free mass), and normed against population references. [116] Studies linking hypermetabolism to disease severity and progression are also warranted.

**Skeletal muscle histology, mtDNA heteroplasmy, and mtDNA density**. Human skeletal muscle from the diaphragm was subjected to sequential cytochrome c oxidase (COX, diaminobenzidine, brown) and succinate dehydrogenase (SDH, nitrobluetetrazolium, blue) staining as described previously[117]. This technique reveals segments of myofibers deficient for mtDNA-encoded COX but positive for exclusively nDNA-encoded SDH[22]. Sub-cellular segments of the same myofiber highlighted in Fig. 1b were dissected from a 20um-thick cryosection by laser-capture microdissection (LCM) on a Leica AS LMD 6000 microscope, transferred and digested (Tween20, Proteinase K) overnight, and used as template DNA in a multiplex real-time PCR reaction that amplifies *MT-ND4* and *MT-ND1* amplicons within the minor and major arcs of the mtDNA, respectively, to calculate heteroplasmy levels for major arc mtDNA deletions[118]. Total mtDNA density was quantified by deriving *MT-ND1* copies from a standard curve, normalized per surface area ($\mu m^2$) of tissue used as input[117].

**Tissue culture**. Primary human dermal fibroblasts were obtained from a distributor or in the local clinic from 3 healthy and 3 SURF1-patient donors (IRB #AAAB0483, see Tables 2 & 3 for descriptive information and distributor). Fibroblasts were isolated from skin tissue biopsies using standard procedures. After isolation, fibroblasts were stored in 10% DMSO (Sigma-Aldrich #D4540), and 90% fetal bovine serum (FBS, Life Technologies #10437036) in a cryogenic tube under liquid nitrogen. To avoid freeze shock, necrosis cells were frozen gradually in an isopropanol container (Thermofisher #5100-0001) at −80 °C overnight before storage in liquid nitrogen.

Genotypes were confirmed by whole genome sequencing. Paired-end (PE) reads were obtained from Illumina HiSeq and processed using SAMtools (v1.2) and BaseSpace workflow (v7.0). PE reads were aligned to hg19 genome reference (UCSC) using Isaac aligner (v04.17.06.15), and BAM files were generated. Small variants, including single nucleotide variants (SNVs) and insertion/deletion (Indels), were called from the entire genome using Strelka germline variant caller (v2.8). Variants specific to the SURF1 gene were obtained from the genome-wide annotated vcf files using SnpSift and annotated using web ANNOVAR.

To initiate cultures, cryopreserved fibroblasts were thawed at 37 °C (< 4 min) and immediately transferred to 20 ml of preheated DMEM (Invitrogen #10567022). Cells were cultured in T175 flasks (Eppendorf #0030712129) at standard 5% $CO_2$ and atmospheric (~21%) $O_2$ at 37 °C in DMEM (5.5 mM glucose) supplemented with 10% FBS, 50 µg/ml uridine (Sigma-Aldrich #U6381), 1% MEM non-essential amino acids (Life Technologies #11140050), 10 µM palmitate (Sigma-Aldrich #P9767) conjugated to 1.7 µM bovine serum albumin (BSA) (Sigma-Aldrich #A8806), and 0.001% DMSO (treatment-matched, Sigma-Aldrich #D4540). Cells were passaged approximately every 5 days (±1 day). Oligo-treated healthy control cells were cultured in the same media as control cells supplemented with 1 nM Oligomycin (in 0.001% DMSO, Sigma-Aldrich #75351) starting on Day 15.

Brightfield microscopy images (10×, 20× magnification) were taken before each passage using an inverted phase-contrast microscope (Fisher Scientific #11350119). Cell counts, volume, and death were determined at each passage using the Countess II Automated Cell Counter (ThermoFisher Scientific #A27977). Growth rates were used to determine replating density, by pre-calculating the number of cells needed to reach ~90% confluency (~2.5 million cells) at time of the next passage. Cells were never plated below 200,000 cells or above 2.5 million cells to avoid plating artifacts of isolation or contact inhibition, respectively. The timing and frequency of time points collected vary by assay, with an average sampling frequency of 15 days[119]. Cell media was collected at each passage. Individual cell lines were terminated after exhibiting less than one population doubling over a 30-day period. The Hayflick limit was determined as the total number of population doublings of a cell line at the point of termination.

**Mycoplasma testing**. Mycoplasma testing was performed according to the manufacturer's instructions (R&D Systems #CUL001B) at the end of the lifespan for each treatment and cell line used. All tests were negative.

**Calculations of energy expenditure and normalization to division rate and cell size**. Bioenergetic parameters were measured using the XFe96 Seahorse extracellular flux analyzer (Agilent), and oxygen consumption rate (OCR), and extracellular acidification rate (pH change) were measured over confluent cell monolayers. Cells were plated for Seahorse measurement every 3 passages (~15 days) with 10–12 wells plated per treatment group. Each well of a seahorse 96-well plate was plated with 20,000 cells and incubated overnight under standard

---

### Table 2 Control and SURF1 donor characteristics.

| Cell Line | Tissue | Genotype | Sex | Age | Passage[a] | Source | Cat # |
|-----------|--------|----------|-----|-----|---------|--------|-------|
| Donor 1 | Dermal breast | Normal | Male | 18 | 1 | Lifeline Cell Technology | FC-0024 Lot # 03099 |
| Donor 2 | Dermal breast | Normal | Female | 18 | 1 | Lifeline Cell Technology | FC-0024 Lot # 00967 |
| Donor 3 | Foreskin | Normal | Male | 0 | 4 | Coriell Institute | AG01439 |
| Patient 1 | Dermal upper-arm skin | SURF1 mutation | Male | 0.25 | 7 | Hirano lab | NA |
| Patient 2 | Dermal upper-arm skin | SURF1 mutation | Male | 11 | 5 | Hirano lab | NA |
| Patient 3 | Dermal upper-arm skin | SURF1 mutation | Female | 9 | 9 | Hirano lab | NA |

[a]Indicates the passage at which cells were obtained before the experiment began.

---

### Table 3 Genotyping results of SURF1 patient-derived fibroblasts.

| Cell line | Surf1 mutation | Exonic function | dbSNP id | Clinical significance[a] |
|-----------|----------------|-----------------|----------|-------------------------|
| Patient 1 | c.518_519del (p.S173Cfs*7) c.845_846del (p.S282Cfs*7) | Frameshift deletion | rs782316919 | Pathogenic \| Pathogenic |
| Patient 2 | c.247_248insCTGC (p.R83Pfs*7) c.574_575insCTGC (p.R192Pfs*7) | Frameshift insertion | rs782289759 | NA |
| | c.C246G (p.T82T) C573G (p.T191T) | Synonymous SNV | rs28715079 | Benign \| Likely Benign |
| | c.313_321del (p.L105_A107del) | Nonframeshift deletion | rs759270179 | NA |
| | c.311_312insA (p.L105Sfs*11) | Frameshift insertion | rs764928653 | NA |
| | c.T280C (p.L94L) | Synonymous SNV | rs28615629 | Benign \| Likely Benign |
| Patient 3 | c.C246G (p.T82T) c.C573G (p.T191T) | Synonymous SNV | rs28715079 | Benign \| Likely Benign |
| | Homozygous c.313_321del (p.L105_A107del) | Nonframeshift deletion | rs759270179 | NA |
| | c.T280C (p.L94L) | Synonymous SNV | rs28615629 | Benign \| Likely Benign |

Results from whole genome sequencing (WGS). *SNV* single nucleotide variant.
[a]Clinical interpretation of genetic variants is based on the ANNOVAR gene annotation pipeline that uses ClinVar database as a primary reference.

growth conditions, following the manufacturer's instructions, including a plate wash with complete Seahorse XF Assay media. The complete XF media contains no pH buffers and was supplemented with 5.5 mM glucose, 1 mM pyruvate, 1 mM glutamine, 50 μg/ml uridine, and 10 μM palmitate conjugated to 1.7 μM BSA. After washing, the plate was incubated in a non-$CO_2$ incubator for one hour to equilibrate temperature and atmospheric gases. The instrument was programmed to assess various respiratory states using the manufacturer's MitoStress Test[120]. Basal respiration, ATP turnover, proton leak, coupling efficiency, maximum respiration rate, respiratory control ratio, spare respiratory capacity, and non-mitochondrial respiration were all determined by the sequential additions of the ATP synthase inhibitor Oligomycin (final concentration: 1 μM), the protonophore uncoupler FCCP (4 μM), and the electron transport chain Complex I and III inhibitors, rotenone and antimycin A (1 μM). The optimal number of cells and concentration for the uncoupler FCCP yielding maximal uncoupled respiration was determined based on a titration performed on healthy fibroblasts.

The final Seahorse injection included Hoechst nuclear fluorescent stain (ThermoFisher Scientific #62249) to allow for automatic cell counting. After each run, cell nuclei were counted automatically using the Cytation1 Cell Imager (BioTek), and raw bioenergetic measurements were normalized to relative cell counts on a per-well basis. ATP metrics were determined using the P/O ratios of OxPhos and glycolysis as previously described by Mookerjee et al.[59]. These conversions assumed energy sourced was derived entirely from glucose. All $J_{ATP}$ measurements take into account non-mitochondrial, and proton leak-derived oxygen consumption, thereby reflecting the mitochondrial ATP-synthesis-related flux (Supplementary Fig. 2a). The code and raw data are available as detailed in the Data Availability statement.

To assess if increased ECAR in experimental conditions was due to non-glycolytic activity, a glucose-dependency test was performed using the Seahorse XF Glycolysis Stress Test Kit (Agilent, 103020-100). Prior to extracellular flux measurements, young healthy control (Donor2) and young SURF1 (Patient3) cells were grown overnight in differing nutrient conditions: physiological 5.5 mM glucose, 0 mM glucose, 25 mM glucose. The glycolysis stress test kit was performed according to the manufacturer's protocol. To monitor growth and cell death, cells were cultured for 7 days in each glucose condition and monitored daily (see Supplementary Fig. 3).

**mtDNA deletions.** mtDNA deletions were initially detected by long-range PCR (LR-PCR) from DNA extracted from cultured fibroblasts using DNeasy blood and tissue kit (Qiagen #69504) following the manufacturer's instructions. Isolated DNA was amplified using 12 F (np 5855–5875) and D2 R (np 129-110) Oligonucleotide primers to yield a 10-kb product. PCR reactions were carried out using Hot Start TaKaRa LA Taq kit (Takara Biotechnology, #RR042A) with the following cycling conditions: 1 cycle of 94 °C for 1 min; 45 cycles of 94 °C for 30 s, 58 °C for 30 s, and 68 °C for 11 min with a final extension of 72 °C for 12 min. Amplified PCR products were separated on 1% agarose gels in 1× TBE buffer, stained with GelGreen (Biotium #41005), and imaged using a GelDoc Go Imager (Biorad). Primers (5′–3′) were: Forward (12F): AGATTTACAGTCCAATGCTTC (nucleotide position 5855–5875); Reverse (D2R): AGATACTGCGACATAGGGTG (129-110).

**mtDNA next-generation sequencing and eKLIPse analysis.** The entire mtDNA was amplified in two overlapping fragments using a combination of mtDNA primers. The primer pairs used for PCR amplicons were tested first on Rho cells devoid of mtDNA to remove nuclear-encoded mitochondrial pseudogene (NUMTS) amplification (PCR1: 5′-AACCAAACCCCAAAGACACC-3′ and 5′-GCCAATAATGACG TGAAGTCC-3′; PCR2: 5′-TCCCACTCCTAAACACATCC-3′ and 5′-TTTATGG GGTGATGTGAGCC-3′). Long-range PCR was performed with the Kapa Long Range DNA polymerase according to the manufacturer's recommendations (Kapa Biosystems, Boston, MA, mtDNA next-generation sequencing and USA), with 0.5 μM of each primer and 20 ng of DNA. The PCR products were analyzed on a 1% agarose gel electrophoresis.

NGS Libraries were generated using an enzymatic DNA fragmentation approach using Ion Xpress Plus Fragment Library Kit. Libraries were diluted at 100 pM before sequencing and pooled by a maximum of 25 samples. Sequencing was performed using an Ion Torrent S5XL platform using Ion 540 chip™. Signal processing and base calling were done by the pre-processing embedded pipeline. Demultiplexed reads were mapped according to the mtDNA reference sequence (NC_012920.1) before being analyzed with a dedicated homemade pipeline, including eKLIPse[121] (https://github.com/dooguypapua/eKLIPse) using the following settings. Deletion counts have been estimated with a variant call cutoff of >5% heteroplasmy, and separately with cutoffs of 1% and 5% heteroplasmy (see Supplementary Fig. 8d).

- Read threshold: min Quality = 20 | min length = 100 bp
- Soft-Clipping threshold: Read threshold: Min soft-clipped length = 25pb | Min mapped Part = 20 bp
- BLAST thresholds: min = 1 | id = 80 | cov = 70 | gapopen = 0 | gapext = 2
- Downsampling: No

*mtDNA copy number.* Cellular mtDNA content was quantified by qPCR on the same genomic material used for other DNA-based measurements. Duplex qPCR reactions with Taqman chemistry were used to simultaneously quantify mitochondrial

(mtDNA, ND1) and nuclear (nDNA, B2M) amplicons, as described previously[4]. The reaction mixture included TaqMan Universal Master mix fast (life technologies #4444964), 300 nM of custom design primers and 100 nM probes: ND1-Fwd: GAGCGATGGTGAGAGCTAAGGT, ND1-Rev:CCCTAAAACCCGCCACATCT, ND1-Probe: HEX-CCATCACCCTCTACATCACCGCCC-3IABkFQ. B2M-Fwd: CCAGCAGAGAATTGGAAAGTCAA, B2M-Rev: TCTCTCTCCATTCTTCAGT AAGTCAACT, B2M-Probe: FAM-ATGTGTCTGGGTTTCATCCATCCGAC A-3IABkFQ. The samples were cycled in a QuantStudio 7 flex qPCR instrument (Applied Biosystems) at 50 °C for 2 min, 95 °C for 20 s, 95 °C for 1 min, 60 °C for 20 s, for 40 cycles. qPCR reactions were set up in triplicates in 384 well qPCR plates using a liquid handling station (epMotion5073, Eppendorf), in volumes of 20 μl (12 μl mastermix, 8 μl template). Triplicate values for each sample were averaged for mtDNA and nDNA. Ct values >33 were discarded. For triplicates with a C.V. > 0.02, the triplicates were individually examined, and outlier values were removed where appropriate (e.g., >2 standard deviations above the mean), with the remaining duplicates, were used. The final cutoff for acceptable values was set at a C.V. = 0.1 (10%); samples with a C.V. > 0.1 were discarded. A standard curve along with positive and negative controls was included on each of the seven plates to assess plate-to-plate variability and ensure that values fell within the instrument range. The final mtDNAcn was derived using the ΔCt method, calculated by subtracting the average mtDNA Ct from the average nDNA Ct. mtDNAcn was calculated as $2^{\Delta Ct} \times 2$ (to account for the diploid nature of the reference nuclear genome), yielding the estimated number of mtDNA copies per cell.

**Cytokines.** Two multiplex fluorescence-based arrays were custom-designed with selected cytokines and chemokines most highly correlated with age in human plasma from[69], listed as available analytes on the R&D custom Luminex arrays (R&D, Luminex Human Discovery Assay (33-Plex) LXSAHM-33 and LXSAHM-15, http://biotechne.com/l/rl/YyZYM7n3). Media samples were collected at selected passages across the cellular lifespan and frozen at −80 °C. After thawing, samples were centrifuged at 500xg for 5 min, and the supernatant was moved to a new tube. Wells were loaded with media samples diluted 1:5 with assay diluent, incubated, washed, and read on a Luminex 200 (Luminex, USA) as per the manufacturer's instructions. Positive (aged healthy fibroblast) and negative controls (fresh untreated media) samples were used in duplicates on each plate to quantify batch variations. Data were fitted, and final values interpolated from a standard curve in xPONENT (v4.2), normalized to the cell number at the time of collection to produce estimates of cytokine production on a per-cell basis. IL-6 and GDF15 measures were repeated using ELISA, according to the manufacturer's instructions (Abcam #ab229434 and R&D #DGD150).

**Media cell-free DNA.** Total cell-free DNA (cf-DNA) was isolated from cell culture media using a previously published automated, high throughput methodology[122]. Quantitative polymerase chain reaction (qPCR): cf-mtDNA and cf-nDNA levels were measured simultaneously by qPCR. Taqman-based duplex qPCR reactions targeted mitochondrial-encoded ND1 and nuclear-encoded B2M sequences as described previously[122,123]. Each gene assay contained two primers and a fluorescent probe and were assembled as a 20× working solution according to the manufacturer's recommendations (Integrated DNA Technologies). The assay sequences are: ND1 forward 5′-GAGCGATGGTGAGAGCTAAGGT-3′, ND1 reverse 5′-CCCTAAAACCCGCCACATCT-3′, ND1 probe 5′-/5HEX/CCATCAC CC/ZEN/TCTACATCACCGCCC/2IABkGQ/-3′, B2M forward 5′-TCTCTCTCC ATTCTTCAGTAAGTCAACT-3′, B2M reverse 5′-CCAGCAGAGAATGGAAA GTCAA-3′, and B2M probe 5′-/56-FAM/ATGTGTCTG/ZEN/GGTTTCATC CATCCGACCA/3IABkFQ/-3′. Each reaction contained 4 μL of 2× Luna Universal qPCR Master Mix (New England Biolabs, cat#M3003E), 0.4 μL of each 20X primer assay, and 3.2 μL of template cf-DNA for a final volume of 8 μL. The qPCR reactions were performed in triplicates using a QuantStudio 5 Real-time PCR System (Thermo Fisher, cat#A34322) using the following thermocycling conditions: 95 °C for 20 s followed by 40 cycles of 95 °C for 1 s, 63 °C for 20 s, and 60 °C for 20 s. Serial dilutions of pooled human placenta DNA were used as a standard curve.

Digital PCR (dPCR): mtDNA and nDNA copy number (copies/μL) of the standard curve used in cf-mtDNA/cf-nDNA assessment were measured separately using singleplex ND1 and B2M assays using a QuantStudio 3 Digital PCR System and associated reagents (Thermo Fisher, cat#A29154) according to the manufacturer's protocol. The values obtained for the standard curve were used to calculate the copy number for the experimental samples. All reactions were performed in duplicate (two chips). Because the same standard curve was used on all plates, its copy number was applied uniformly to all qPCR plates.

**RNA sequencing and transcriptomic analyses.** Total genomic RNA was isolated every ~11 days across the cellular lifespan and stored in 1 ml TRIzol (Invitrogen #15596026). RNA was extracted on-column using the RNeasy kit (Qiagen #74104), DNase was treated according to the manufacturer's instructions, and quantified using the QUBIT high sensitivity kit (Thermo Fisher Scientific #Q32852). RNA samples underwent QC on a bioanalyzer and Nanodrop 2000; all samples had a RIN score >8.0 and no detectable levels of DNA. RNA (1500 ng/sample, 50 ng/μl) was then submitted for sequencing at Genewiz Inc. (Illumina HiSeq, single index,

10 samples/lane), and underwent RiboZero Gold purification. Sequenced reads yielding approximately 40 million paired-end 150 bp single-end reads per sample. Sequenced reads were then aligned using the pseudoalignment tool, kallisto (v0.44.0)[124]. These data were imported using txi import ('tximport', v1.18.0, length-scaled TPM), and vst normalized ('DEseq2', v1.30.1).

Dimensionality reduction was performed using 'Rtsne' (v0.15) with perplexity value of 10 and initial dimensions of 30 on the log2 transformed normalized expression values after removing genes without any variation in expression across all samples. Linear mixed modeling was performed using the 'lme4' (v1.1) R package with the fixed effects of time grown and clinical group for SURF1-differential expression and fixed effects of time grown and treatment with a mixed effect of the cell line for Oligo-differential expression. P values were obtained by running an analysis of variance (ANOVA) comparing the model for each gene to a null model that had a fixed effect of days grown (mixed effects of cell line for Oligo models), and then the value was corrected for multiple comparisons using FDR-adjustment ($p < 0.05$). We used iPAGE to discover perturbed pathways in SURF1 and Oligo-treated cells (https://tavazoielab.c2b2.columbia.edu/iPAGE/)[77]. iPAGE enables the systematic and comprehensive discovery of pathways that are significantly informative of gene expression measurements without any explicit thresholding requirements. Additionally, iPAGE is also able to detect pathways whose constituent genes are both upregulated and downregulated in the treatments. Input to iPAGE included gene symbols and, for each gene, a cluster identifier indicating if it was upregulated, downregulated, or not differentially expressed in both (i.e., intersection) SURF1-mutant and Oligo-treated cells compared to controls. For discovering significantly over- and under-represented pathways using iPAGE, we used a stringent $p$ value cutoff of 0.001 along with minr=1, ind=0 (to produce the most expansive set of pathway terms) and, catMin=30 (to exclude pathways with fewer than 30 genes). Timecourse and heatmaps show transcript levels relative to the median of the youngest control timepoints. Categorized genes were selected based on known mitochondrial and aging literature. Categorized pathways were categorized into meta-categories based on shared gene ontology parent processes.

**DNA methylation and methylome analysis**. Global DNA methylation was measured using the Illumina EPIC microarray ran at the UCLA Neuroscience Genomic Core (UNGC). DNA was extracted using the DNeasy kit (Qiagen cat#69506) according to the manufacturer's protocol and quantified using the QUBIT broad-range kit (Thermo Fisher Scientific cat#Q32852). At least 375 ng of DNA was submitted in 30 μl of ddH$_2$O to UNGC for bisulfite conversion and hybridization using the Infinium Methylation EPIC BeadChip kit. Sample positions across plates were randomized to avoid batch variation effects on group or time-based comparisons. All DNA methylation data were processed in R (v4.0.2), using the 'minfi' package (v1.36.0). Quality control preprocessing was applied by checking for correct sex prediction, probe quality, and sample intensities, and excluding SNPs and non-CpG probes. Data were then normalized using Functional Normalization. Using the R package 'sva' (v3.12.0), both RCP and ComBat adjustments were applied to correct for probe type and plate bias, respectively. After quality control, DNAm levels were quantified for 865,817 CpG Sites.

Dimensionality reduction was performed using the 'Rtsne' package (v0.15) with a perplexity value of 10 and initial dimensions of 30 on the normalized beta values. We ran LMER using 'lme4' (v1.1). For our differential methylation analysis of SURF1, the fixed effects were assigned to '*days_grown*' and '*clinical_group*'. For the Oligomycin treatment, the fixed effects '*time_grown*' and '*treatment*' and the mixed effect were assigned to the '*cell_lines*' (i.e., donors). P values were obtained from an ANOVA comparing the model for each CpG to a null model with a fixed effect of days grown (mixed effects of the cell line for Oligomycin models) and then corrected for multiple comparisons using FDR-adjustment ($p < 0.05$) to identify differentially methylated CpGs (DMPs). Differentially methylated regions (DMRs) were derived using the modified comb-p method in the 'Enmix' package (v1.26.8), with a maximum distance for DMR combination of 1000 bp, a bin size for autocorrelation of 310, and FDR-adjustment cutoff of 0.01, and a minimum of 3 CpGs per a DMR. Each DMP and DMR were assigned to the nearest annotated gene (IlluminaHumanMethylationEPICanno.ilm10b4.hg19 package, v0.6.0). Gene set enrichment analysis was then performed using ShinyGO[125] (v0.66, http://bioinformatics.sdstate.edu/go/) on the top 1000 DMPs- or DMRs-associated genes based on the combined negative log $p$ value across hyper- and hypo-methylated DMPs and DMRs.

**Relative telomere length**. Relative telomere length was measured by quantitative polymerase chain reaction (qPCR), expressed as the ratio of telomere to single-copy gene abundance (T/S ratio). The telomere length measurement assay was adapted from the published original method by Cawthon[126,127]. The telomere thermal cycling profile consisted of Cycling for T(celomic) PCR: Denature at 96 °C for 1 min, one cycle; denature at 96 °C for 1 s, anneal/extend at 54 °C for 60 s, with fluorescence data collection, 30 cycles. Cycling for S (single copy gene) PCR: Denature at 96 °C for 1 min, one cycle; denature at 95 °C for 15 s, annealing at 58 °C for 1 s, extend at 72 °C for 20 s, 8 cycles; followed by denature at 96 °C for 1 s, annealing at 58 °C for 1 s, extend at 72 °C for 20 s, hold at 83 °C for 5 s with data collection, 35 cycles. The primers for the telomere PCR are tel1b [5′-CGGTTT (GTTTGG)$_5$GTT-3′], used at a final concentration of 100 nM, and tel2b [5′-

GGCTTG(CCTTAC)$_5$CCT-3′], used at a final concentration of 900 nM. The primers for the single-copy gene (human beta-globin) PCR are hbg1 [5′ GCTT CTGACACAACTGTGTTCACTAGC-3′], used at a final concentration of 300 nM, and hbg2 [5′-CACCAACTTCATCCACGTTCACC-3′], used at a final concentration of 700 nM. The final reaction mix contained 20 mM Tris-HCl, pH 8.4; 50 mM KCl; 200 μM each dNTP; 1% DMSO; 0.4× SYBR Green I; 22 ng E. coli DNA; 0.4 Units of Platinum Taq DNA polymerase (Invitrogen Inc.); approximately 6.6 ng of genomic DNA per 11 microliter reaction. Tubes containing 26, 8.75, 2.9, 0.97, 0.324, and 0.108 ng of a reference DNA (Human genomic DNA from the buffy coat, Sigma cat# 11691112001) are included in each PCR run so that the number of targeted templates in each research sample can be determined relative to the reference DNA sample by the standard curve method. The same reference DNA was used for all PCR runs. Assays were run in triplicate wells on 384-well assay plates in a Roche LightCycler 480. The average concentrations of T and S from the triplicate wells were used to calculate the T/S ratios after a Dixon's Q test to remove outlier wells from the triplicates. The *T/S* ratio for each sample was measured twice. When the duplicate T/S value and the initial value varied by more than 7%, the sample was run the third time, and the two closest values were reported. 26 out of the 512 samples (5%) have a CV greater than 10% after the third measurement. The inter-assay coefficient of variation (CV) for this study is 3.0% ± 4.3% (including the 26 samples) and 2.2% ± 2.0% (excluding the 26 samples). Telomere length assays for the entire study were performed using the same lots of reagents. Lab personnel lab who performed the assays were provided with de-identified samples and were blind to other data.

**DNAmAge**. DNAmAge was calculated using the online calculator (https://dnamage.genetics.ucla.edu/new) with normalization using the age of the cell line donor as the input age. This outputted the Horvath1 (i.e., PanTissue clock), Horvath2 (Skin&Blood clock), PhenoAge, Hannum, and GrimAge estimated DNAmAges. PC-based DNAmAges were then obtained using the principal component method (https://github.com/MorganLevineLab/PC-Clocks)[86]. The rates of epigenetic aging for each cell line were determined from the linear slope of timepoints between 25 and 75 days. This period ensures that Oligo treatment has taken effect, and avoids late-life changes in the behavior of DNAm clocks, providing the time window where the signal is most stable.

**Statistics and reproducibility**. All statistical analyses were performed using GraphPad Prism (v9.0) and RStudio (v1.3.1056) using R (v4.0.2). Comparisons of groups between control, SURF1, and treatment groups were performed using a mixed effects model, except for peak and rate measurements (unpaired *t*-test, assuming unequal variance or two-way ANOVA for concurrent measures). Interpolated curves for each experimental group are the best fit for non-linear third-order or fifth-order polynomial functions depending on the kinetic complexity of a given measurement. Data visualization and statistical analyses were generated in R ('ggplot2', v3.3.5) and Prism 8.

The time windows for specific statistical analyses were selected based on a combination of cellular growth behavior including (i) population doubling curves (e.g., stable division rates for all groups early in the cellular lifespan between days 20 and 50), (ii) the availability of matching timepoints between treatment groups (at least three timepoints for all groups), and (iii) potential delay to reach stable cellular phenotypes in Oligo-treated cells. To allow for adjustment to the in vitro environment, treatments began after 15 days of culture. Therefore, overall "lifespan effects" were determined between 20–150 days, which represents the maximal replicative lifespan of SURF1 cells. "Early life" effects that isolate most clearly the effects of OxPhos dysfunction, and avoid the potential accelerated aging phenotypes in SURF1 and Oligo cells, were examined between days 20 and 50 days. For analyses of differentially expressed genes (RNASeq, Fig. 6) and differential methylation (DNAm, Fig. 7) where a greater datapoint density was necessary to achieve robust mixed effects models, SURF1 cells were analyzed between 0 and 75 days (genetic defects in *SURF1* are constitutive so do not require time in culture to manifest) whereas models for Oligo-treated cells used timepoints between days 35 to 110 (allowing 15 days for the effects of ATP synthase inhibition to manifest in the transcriptome, while avoiding late-life changes). All timepoints are shown in time series graphs.

**Reporting summary**. Further information on research design is available in the Nature Portfolio Reporting Summary linked to this article.

## Data availability

The RNAseq and DNA methylation datasets for this project are available under the GEO SuperSeries GSE179849. All data preprocessing and analysis code is available on GitHub (https://github.com/gav-sturm/Cellular_Lifespan_Study). Data presented in this manuscript was generated as part of the Cellular Lifespan Study, which includes metabolic and endocrine experimental treatments across multiple donors described in detail in[119]. The complete multi-omic fibroblast dataset for the present study is available without restriction and can be accessed, visualized, and downloaded using our web tool: https://columbia-picard.shinyapps.io/shinyapp-Lifespan_Study/. This fibroblast dataset,

along with brightfield images and seahorse assay files, are additionally available at FigShare.com under accession numbers 18441998, 18444731, 20277606, respectively. The meta-analyzed clinical data of mitochondrial disease cohorts (Fig. 1) can be obtained from the original publications listed in Table 1. Requests for any other information will be provided upon request by the corresponding author.

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

## Acknowledgements

We are grateful to Jane Newman, Renae Stefanetti, Robert W Taylor, and Gráinne S Gorman (Wellcome Center for Mitochondrial Research) for contributing data for Cohort 2, Rohit Sharma and Vamsi Mootha (Massachusetts General Hospital) for contributing data for Cohort 4, Robert McFarland (Wellcome Center for Mitochondrial Research) for contributing data for Cohort 17, and other investigators whose work contributed to the meta-analysis in Fig. 1. We thank Marlon McGill for technical assistance with parts of this project and Herman Pontzer for analytical advice. The cellular studies and analyses were supported by NIH grant R01AG066828 and the Baszucki Brain Research Fund to M.P., the J. Willard and Alice S. Marriott Foundation, Muscular Dystrophy Association, Nicholas Nunno Foundation, JDF Fund for Mitochondrial Research, and Shuman Mitochondrial Disease Fund to M.H. Further support was provided by NIH grants R01GM123771 and R35HL155670 awarded to E.L.S. and M-P.S-O., respectively. All research at Great Ormond Street Hospital NHS Foundation Trust and UCL Great Ormond Street Institute of Child Health is made possible by the NIHR Great Ormond Street Hospital Biomedical Research Centre. The views expressed are those of the author(s) and not necessarily those of the NIH, NHS, the NIHR, or the Department of Health.

## Author contributions

G.S. and M.P. designed experiments. M.H. provided cell lines. G.S. performed cellular studies, and processed samples with assistance from A.S.M. and A.T. A.S.M. performed replication and no-glucose experiments. G.S. analyzed data with assistance from A.S.M., B.S., and A.C. K.R.K. and performed long-range PCR, cytokine arrays, and WGS analysis. S.W. and B.K. measured cf-mtDNA. J.L. and E.S.E. measured telomere length. A.H.C., M.L., and S.H. contributed epigenetic clocks. B.S. and S.T. performed the iPAGE analyses. C.B., V.P., and G.L. performed mtDNA sequencing. T.T., S.R., B.G., R.G.H., and J.V. provided data for the meta-analysis of clinical data. M.P. and M.C. performed a meta-analysis. M.J.M., D.C.W., and M-P.S-O. contributed to the interpretation of findings. M.P., G.S., M.H., and E.L.S. drafted the paper. All authors reviewed the final version of the paper.

## Competing interests

The authors declare no competing interests.

## Additional information

[1]Department of Psychiatry, Division of Behavioral Medicine, Columbia University Irving Medical Center, New York, NY, USA. [2]Department of Biochemistry and Biophysics, University of California, San Francisco, CA, USA. [3]Departments of Biological Sciences, Systems Biology, and Biochemistry and Molecular Biophysics, Institute for Cancer Dynamics, Columbia University, New York, NY, USA. [4]Department of Physiology and Functional Genomics, Clinical and Translational Research Building, University of Florida, Gainesville, FL, USA. [5]Department of Genetics and Neurology, Angers Hospital, Angers, France. [6]UMR CNRS 6015, INSERM U1083, MITOVASC, SFR ICAT, Université d'Angers, Angers, France. [7]Department of Medicine, Vascular Medicine Institute and Center for Metabolic and Mitochondrial Medicine, University of Pittsburgh, Pittsburgh, PA, USA. [8]Internal Medicine-Pediatrics Residency Program, University of Pittsburgh Medical Centre, Pittsburgh, PA, USA. [9]Department of Psychiatry, Yale University School of Medicine, New Haven, CT, USA. [10]Department of Anesthesiology and Critical Care Medicine, The Children's Hospital of Philadelphia, Philadelphia, PA, USA. [11]Center for Mitochondrial and Epigenomic Medicine, The Children's Hospital of Philadelphia, Philadelphia, PA, USA. [12]Department of Epidemiology and Population Health, Stanford University, Stanford, CA, USA. [13]Department of Psychiatry and Behavioral Sciences, University of California, San Francisco, CA, USA. [14]Mitochondrial Research Group, UCL Great Ormond Street Institute of Child Health, and Metabolic Unit, Great Ormond Street Hospital for Children NHS Foundation Trust, London, UK. [15]Copenhagen Neuromuscular Center, Department of Neurology, Rigshospitalet, University of Copenhagen, Copenhagen, Denmark. [16]Department of Medicine, University of Udine, Udine, Italy. [17]Altos Labs, San Diego, CA, USA. [18]Neuromuscular Center, Institute for Exercise and Environmental Medicine of Texas Health Resources and Department of Neurology, University of Texas Southwestern Medical Center, Dallas, TX, USA. [19]Center of Excellence for Sleep & Circadian Research and Division of General Medicine, Department of Medicine, Columbia University Irving Medical Center, New York, NY, USA. [20]Department of Pathology and Genomic Medicine, and MitoCare Center, Thomas Jefferson University, Philadelphia, PA, USA. [21]Department of Neurology, H. Houston Merritt Center, Columbia Translational Neuroscience Initiative, Columbia University Irving Medical Center, New York, NY, USA. [22]New York State Psychiatric Institute, New York, NY, USA. ✉email: martin.picard@columbia.edu

