## [Peer Review File · Communications Biology]

This manuscript has been previously reviewed at another Nature Portfolio journal. This document only contains reviewer comments and rebuttal letters for versions considered at Communications Biology.

REVIEWERS' COMMENTS:

Reviewer #1 (Remarks to the Author):

The authors have addressed my prior concerns. No further comments.

Reviewer #2 (Remarks to the Author):

I have carefully reviewed the manuscript and reviewers questions (reviewer 1, point 3 and reviewer 3, point 2). I found the manuscript nicely written with original and significant data.

Considering reviewer 1. point 3.

I agree with the authors justifying the use of 5.5 mM glucose, since glucose level range in fasted human blood is between 3.9 and 6 mM. Also many commercial media resembling human plasma such as HPLM contains glucose concentration ranging between 5 to 5.5 mM. I would be careful with authors' response, where they state using 100 mM palmitate, while in the paper they claim using 10 μ M palmitate. The palmitate level should range between 0.3 to 4.1 mM in human blood. This is the only minor comment, since in the presence of 5.5 mM glucose, cells don't utilize fatty acids for respiration.

Considering reviewer 3 point 2.

As stated in my first response, the 5.5 mM glucose is not "high level glucose" and fits to the physiological plasma glucose range. Therefore, cells should use their OXPHOS system to generate ATP. The authors confirmed that the majority of energy comes from OXPHOS and only 36% was generated by glycolysis. Authors also added Extended data to figure 5 responding to lower oxygen. Obviously, the fibroblasts under 3% oxygen behaved in a slightly different way, but still confirming the original hypothesis. Nevertheless, the telomere shortening and de novo mtDNA deletions formation were not measured since authors did only 40 days of cultivation. As the reviewer 3 proposed, one may speculate, if the hypermetabolism is a direct cause of telomere shortening and mtDNA deletions or if this is due to a specific stress related to mitochondrial deficiency. However, most of the laboratories cultivate cells under atmospheric oxygen tension. Therefore, if using proper controls it must be generally accepted as a correct approach. I would suggest to authors to raise the issue of possible impact of mitochondrial deficiency and oxygen tension on DNA biology in discussion. One must not forget that surf1 is an assembly protein of complex IV and therefore the oxygen level must be taken critically.

In conclusion, the paper is quite dense and brings new discoveries worth publishing in Communications Biology.

COMMSBIO-22-3050-T
Communications Biology
Response to reviewers

Reviewer #2:

I have carefully reviewed the manuscript and reviewers questions (reviewer 1, point 3 and reviewer 3, point 2). I found the manuscript nicely written with original and significant data.

Considering reviewer 1. point 3.

I agree with the authors justifying the use of 5.5 mM glucose, since glucose level range in fasted human blood is between 3.9 and 6 mM. Also, many commercial media resembling human plasma such as HPLM contains glucose concentration ranging between 5 to 5.5 mM. I would be careful with authors' response, where they state using 100 mM palmitate, while in the paper they claim using 10 μ M palmitate. The palmitate level should range between 0.3 to 4.1 mM in human blood. This is the only minor comment, since in the presence of 5.5 mM glucose, cells don't utilize fatty acids for respiration.

We thank the reviewer for these positive and constructive criticisms. As mentioned, the 100 mM palmitate described in our response was a typo. Supplemental palmitate was used at a concentration of 10 μ M throughout the study. We have carefully reviewed all materials and confirm that the 10 μ M concentration is reported correctly throughout the manuscript.

Considering reviewer 3 point 2.

As stated in my first response, the 5.5 mM glucose is not "high level glucose" and fits to the physiological plasma glucose range. Therefore, cells should use their OXPHOS system to generate ATP. The authors confirmed that the majority of energy comes from OXPHOS and only 36% was generated by glycolysis. Authors also added Extended data to figure 5 responding to lower oxygen. Obviously, the fibroblasts under 3% oxygen behaved in a slightly different way, but still confirming the original hypothesis. Nevertheless, the telomere shortening and de novo mtDNA deletions formation were not measured since authors did only 40 days of cultivation. As the reviewer 3 proposed, one may speculate, if the hypermetabolism is a direct cause of telomere shortening and mtDNA deletions or if this is due to a specific stress related to mitochondrial deficiency. However, most of the laboratories cultivate cells under atmospheric oxygen tension. Therefore, if using proper controls it must be generally accepted as a correct approach. I would suggest to authors to raise the issue of possible impact of mitochondrial deficiency and oxygen tension on DNA biology in discussion. One must not forget that surf1 is an assembly protein of complex IV and therefore the oxygen level must be taken critically. In conclusion, the paper is quite dense and brings new discoveries worth publishing in Communications Biology.

Thank you for this added nuance. We have added a discussion of the potential influence of oxygen tension on page 26: "Oxygen tension affecting electron flux through the OxPhos system could also contribute to impose energetic tradeoffs. However, our cells grown at 3% oxygen did not show significantly different growth rates nor energy expenditure. On this

basis, we can largely rule out a main effect of oxygen tension in our results and instead conclude that hypermetabolic cellular phenotype is a direct consequence of the OxPhos defects. This interpretation is reinforced by the orthogonal nature of the SURF1 (complex IV) and Oligo (complex V) models, which target different OxPhos components yet produce comparable hypermetabolism.”